# A self-healing plastic ceramic electrolyte by an aprotic dynamic polymer network for lithium metal batteries

Yubin He[1,7], Chunyang Wang [1,7], Rui Zhang[1], Peichao Zou[1], Zhouyi Chen[1], Seong-Min Bak[2,6], Stephen E. Trask [3], Yonghua Du [2], Ruoqian Lin[4], Enyuan Hu [5] & Huolin L. Xin [1]✉

Oxide ceramic electrolytes (OCEs) have great potential for solid-state lithium metal ($Li^0$) battery applications because, in theory, their high elastic modulus provides better resistance to $Li^0$ dendrite growth. However, in practice, OCEs can hardly survive critical current densities higher than $1 \, mA/cm^2$. Key issues that contribute to the breakdown of OCEs include $Li^0$ penetration promoted by grain boundaries (GBs), uncontrolled side reactions at electrode-OCE interfaces, and, equally importantly, defects evolution (e.g., void growth and crack propagation) that leads to local current concentration and mechanical failure inside and on OCEs. Here, taking advantage of a dynamically cross-linked aprotic polymer with non-covalent $-CH_3 \cdots CF_3$ bonds, we developed a plastic ceramic electrolyte (PCE) by hybridizing the polymer framework with ionically conductive ceramics. Using in-situ synchrotron X-ray technique and Cryogenic transmission electron microscopy (Cryo-TEM), we uncover that the PCE exhibits self-healing/repairing capability through a two-step dynamic defects removal mechanism. This significantly suppresses the generation of hotspots for $Li^0$ penetration and chemomechanical degradations, resulting in durability beyond 2000 hours in $Li^0$-$Li^0$ cells at $1 \, mA/cm^2$. Furthermore, by introducing a polyacrylate buffer layer between PCE and $Li^0$-anode, long cycle life >3600 cycles was achieved when paired with a 4.2 V zero-strain cathode, all under near-zero stack pressure.

Solid-state electrolytes (SSE) potentially offer higher energy density, better resistance to $Li^0$ dendrites, and enhanced safety compared with conventional flammable, volatile, and leakable liquid electrolytes, and therefore are gaining increasing interest for applications in electric vehicles and large-scale energy storage systems[1]. Among all types of SSE, oxide-based ceramic electrolytes (OCEs) possess advantages such as high elastic modulus, better electrochemical stability than sulfide-based SSE, low cost, and environment-benign properties[2]. However, their wide application in solid-state $Li^0$ batteries (SSLMB) was prevented by a number of entangled chemical, electrochemical, and mechanical challenges: (1) The ionic conductivity of OCEs is primarily limited by sluggish $Li^+$ diffusion kinetics through grain boundaries[3,4]. (2) The high electronic conductivity of OCEs can easily result in the direct deposit of $Li^0$ dendrites at grain boundaries[5]. Defects and cracks

[1]Department of Physics and Astronomy, University of California, Irvine, CA, USA. [2]National Synchrotron Light Source II, Brookhaven National Laboratory, Upton, NY, USA. [3]Cell Analysis, Modeling, and Prototyping Facility, Argonne National Laboratory, Lemont, IL, USA. [4]Department of Mechanical Engineering, University of California, Riverside, CA, USA. [5]Chemistry Division, Brookhaven National Laboratory, Upton, NY, USA. [6]Present address: Department of Materials Science and Engineering, Yonsei University, Seoul 03722, Republic of Korea. [7]These authors contributed equally: Yubin He, Chunyang Wang. ✉e-mail: huolin.xin@uci.edu

generated during OCE fabrication and battery cycling are considered "hotspots" for dendrite formation[6,7]. (3) The lithiation/degradation of OCE in contact with $Li^0$ will lead to uncontrolled SEI growth[8,9]. The insufficient wettability of $Li^0$ and the high modulus of OCEs result in high contact resistance and electrode-electrolyte delamination, especially when operating under high current density and large areal capacity[10]. (4) The hot-press sintering process at high pressure/temperature complicates the fabrication process, and the OCEs' brittleness is non-compatible with conventional roll-to-roll battery fabrication techniques[1]. Due to the above drawbacks, current OCEs normally exhibit critical current density as low as $<1\,mA/cm^2$, small operating areal capacity ($0.2\,mAh/cm^2$), high stack pressure requirement ($>40\,MPa$), and poor durability evidenced by short-circuiting or rapid overpotential building-up within a few hundred hours[7,11–14]. Existing strategies like optimizing the sintering conditions to reduce impurities and defects at grain boundaries[15], employing alloy anode (negative electrode) such as LiIn, NaK, LiGa, etc. to avoid dendrite formation[16,17], and introducing artificial SEI to stabilize SSE-electrode interfaces[18] have led to steady progress in performance, such as approaching a high critical current density of $3.2\,mA/cm^2$ and enhanced full cell durability >100 cycles[15,19]. However, successfully implementing the SSLMB technology requires simultaneously addressing all the fundamental challenges of conductivity, dendrite growth, interphase, stack pressure, and fabrication problems.

Here, to tackle the above challenges, we report a plastic ceramic electrolyte (PCE) by embedding a commercial $Li_{1.5}Al_{0.5}Ti_{1.5}(PO_4)_3$ (LATP, 70 wt%) powder into a self-healing solid polymer electrolyte (SH-SPE, 30 wt%) with aprotic dynamic bonding network (Fig. 1a). As evidence, the magic angle spinning solid-state NMR (MAS-ssNMR) spectra presented in Fig. 1b and Supplementary Fig. S1 demonstrate that as the (trifluoromethane) sulfonimide lithium methacrylate (MTFSI) content increases, the signals of ethylene acrylate (EA) shift gradually downfield. This shift can be attributed to the electron-withdrawing effects of the F/O atoms in MTFSI. Conversely, with increasing EA content, the signals of MTFSI shift upfield (Supplementary Fig. S1c), which further supports the presence of non-covalent interactions between EA and MTFSI. Previous theoretical predictions also show a high binding energy of 0.4–0.5 eV for this $-CH_3\cdots CF_3$ interaction[20,21], which surpasses the strength of water-water hydrogen bonding (0.25 eV)[22]. Consequently, this non-covalent interaction is extensively utilized in developing functional polymers, including mechanically robust ionogels[23] and stretchable elastomers[24]. In this work, this non-covalent nature of $-CH_3\cdots CF_3$ interaction leads to a dynamically and reversibly crosslinked network[24,25], where the breaking and reconnection of $-CH_3\cdots CF_3$ bonding enables adaptive migration of ceramic particles inside the polymer matrix (evidence will be discussed later), which has never been achieved in conventional hybrid SSEs where the ceramic is immobilized by crystalized or crosslinked polymer chains with no self-healing capability. In addition, the aprotic $-CH_3$ group in ethylene acrylate monomer contains no reactive hydrogen like in hydroxyl (O–H)[26] and amine (N–H) groups[27], which also helps avoid the side reaction with $Li^0$-anode.

A cold-milling strategy was employed to prepare the PCE with desirable free-standing, flexible, and deformable properties (Fig. 1c) for potential roll-to-roll SSLMB fabrication and low stack pressure operation (<0.1 MPa). The PCE also demonstrates remarkable self-healing capabilities which make long-life SSLMBs possible (Fig. 1c). For the first time, the healing kinetics of millimeter-sized cracks/defects was directly visualized by *operando* X-ray fluorescence (XRF) microscopy. Cryo-TEM chemical analysis reveals that grain boundaries of LATP can be well infiltrated and protected by SH-SPE (Fig. 1d), leading to a 33-fold increase in grain boundary ion conductivity (0.8 mS/cm *vs.* 0.024 mS/cm) and a 32-fold decrease in electron conductivity (5.7E−8 S/cm *vs.* 1.5E−6 S/cm at 30 °C). Cryo-TEM characterization further revealed densely packed, dome-shaped $Li^0$ deposits well-protected by a uniform and compact

SPE-derived SEI layer. Owing to the uncommon self-healing capability, optimized ion/electron conduction at the grain boundary, and improved interfacial stability, the PCE-based $Li^0$-$Li^0$ cell delivered dendrites-free cycling for ~2000 h at 22 °C and 1 mA/cm² (Fig. 1e, enlarged voltage-time curves showed in Supplementary Fig. S2). To further eliminate the possible side reaction of LATP after long cycles (e.g., >2000 h), we developed a hierarchical SSE (H-SSE) with a polyacrylate-based SPE as a buffer layer between PCE and the $Li^0$-anode (see Methods for details of the buffer layer). When paired with a 4.2 V high-Ni zero-Co zero-strain cathode (positive electrode), the H-SSE-based full cell operated stably for 3600 cycles without short-circuiting. Also, remarkably, when paired with a commercial high-loading NMC811 cathode (1.6 mAh/cm²), it shows capacity retention of 71% after 1000 cycles at 22 °C. All results were measured in coin cells with a stack pressure lower than 0.1 MPa.

## Results and discussion

### Self-healing mechanism and interfacial chemistry

The self-healing mechanism of PCE in real SSLMB was investigated by *operando* XRF imaging. Incident X-ray energy of 3000 eV was employed to excite both the sulfur element in SPE (highlighted in green) and the phosphorus element in LATP (highlighted in red) (Fig. 2a). In-situ cells with $Li^0$ as the reference electrode, PCE as the electrolyte, and stainless steel (SS) as the working electrode were assembled inside a Kapton® tube and sealed with epoxy resin to avoid air exposure (Fig. 2b). Note that a thicker PCE of ~2 mm was employed to provide a large volume size for better observation of the self-healing process. While in other electrochemical measurements, the thickness of PCE is 350 μm (Supplementary Fig. S3). Figure 2a highlights a millimeter-sized void deficient of S and P (region R1 with dark contrast) which was naturally formed during battery fabrication (void formation is commonly observed in OCE-based batteries[10]). After cycling at 0.2 mA/cm² for 12 h at 22 °C, the void was largely healed through migration of both SH-SPE and LATP. The self-healed region around the residual void shows evident enrichment of S (green colored in R3 and R4 regions) which is a fingerprint of SH-SPE. In contrast, the completely healed region R4 (close to the bulk PCE) is composed of both SPE and LATP. The observed different healing states in the above indicate a two-step self-healing mechanism: first, SPE infiltrates into the void, and subsequently, micron-sized LATP particles migrate through the SPE matrix to fill the voids.

Next, more detailed in-situ monitoring was performed to further confirm the two-step self-healing mechanism and to understand the healing kinetics. Figure 2c presents time-resolved high-resolution XRF images showing the healing kinetics of two neighboring voids. The two 300-μm-sized voids were completely self-repaired within 20 h. The observed uneven distribution of LATP and SH-SPE after 10 h resulted from the infiltration of SH-SPE and the diffusion of LATP, indicating that the SH-SPE infiltration and LATP diffusion is a dynamic process. Additionally, the varying LATP content in different regions could create a concentration gradient, promoting the diffusion of LATP to low concentration regions. Therefore, with extended battery cycling, we believe this inhomogeneity may gradually decrease and will not significantly affect the dendrite inhibition capability of PCE, as evidenced by the excellent durability of the $Li^0$|PCE|$Li^0$ cell (Fig. 1e). Figure 2d shows the quantitative analysis of the void size evolution with cycling time (the void size is defined as the square root of the void area). It is interesting to find that the self-healing process significantly accelerated with decreased void size. For example, with the void size decreased from ~282 to ~226 μm, the self-healing rate increases from ~5.6 to ~22.6 μm/hour, which is much faster than the $Li^0$-deposition speed at 1 mA/cm² (4.82 μm/hour). This accelerated self-healing rate can be attributed to the three-dimensional nature of voids, where the volume of a void is proportional to the cube of its diameter. Consequently, the self-healing process is significantly accelerated at smaller void sizes due to the much smaller volume.

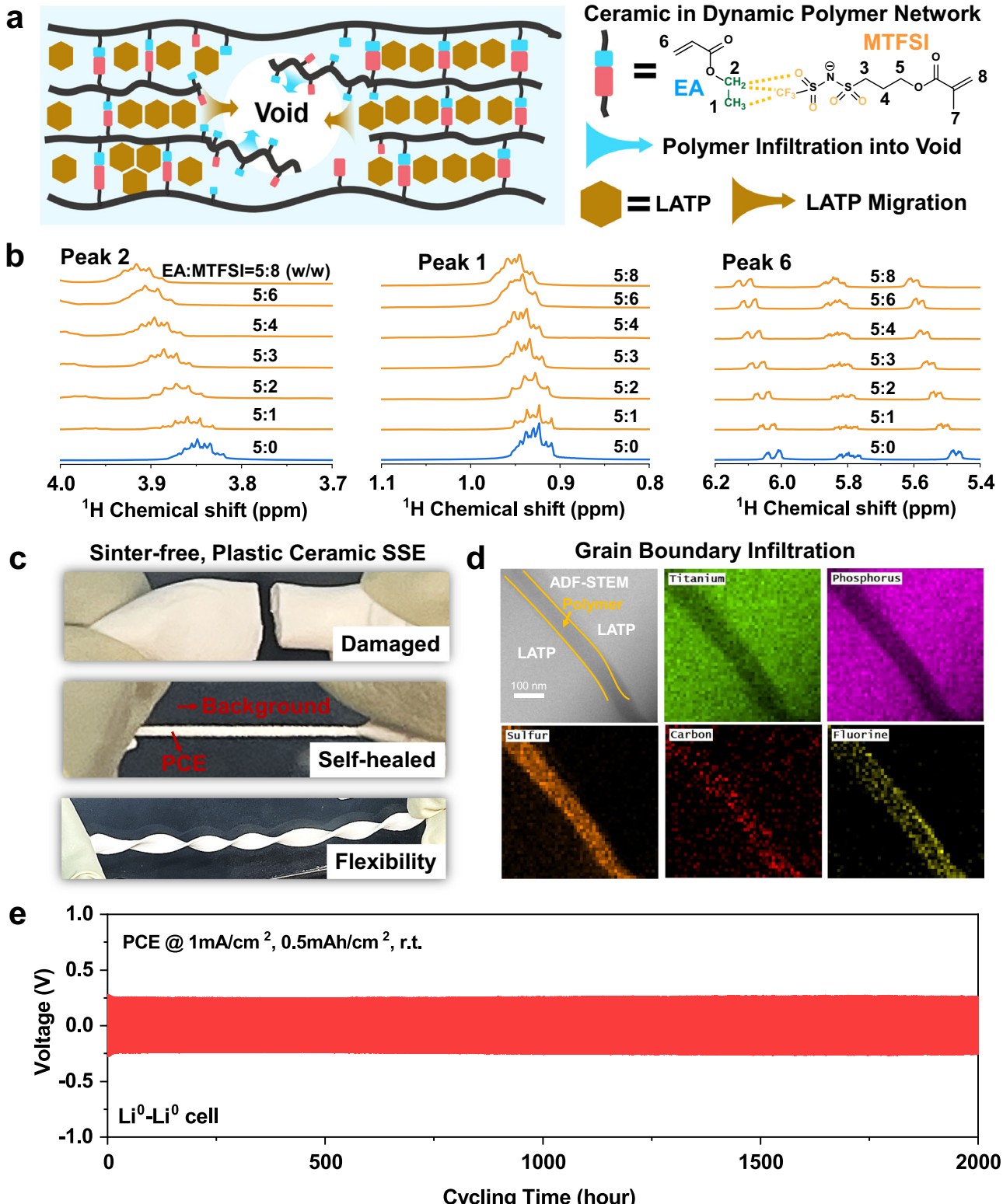

**Fig. 1 | Design concept of the plastic ceramic electrolyte (PCE). a** Schematic illustration showing the design concept of PCE through embedding the LATP powder into a dynamic polymer network. **b** ¹H NMR spectra showing the −CH₃···CF₃ non-covalent interaction between EA and MTFSI monomers. **c** Photographs showing the self-healing ability and flexibility of PCE at 22 °C. After cracking, as shown in the top picture, the PCE can self-heal after hand-milling at room temperature for 1 min, as shown in the middle picture. **d** Cryo-TEM images and EDS mapping of PCE. Enriched C, F, and S elements between two LATP grains show that the grain boundaries were well-infiltrated by the SH-SPE. **e** Voltage-time profiles at 22 °C of the solid-state Li⁰-Li⁰ cells employing the PCE as the electrolyte.

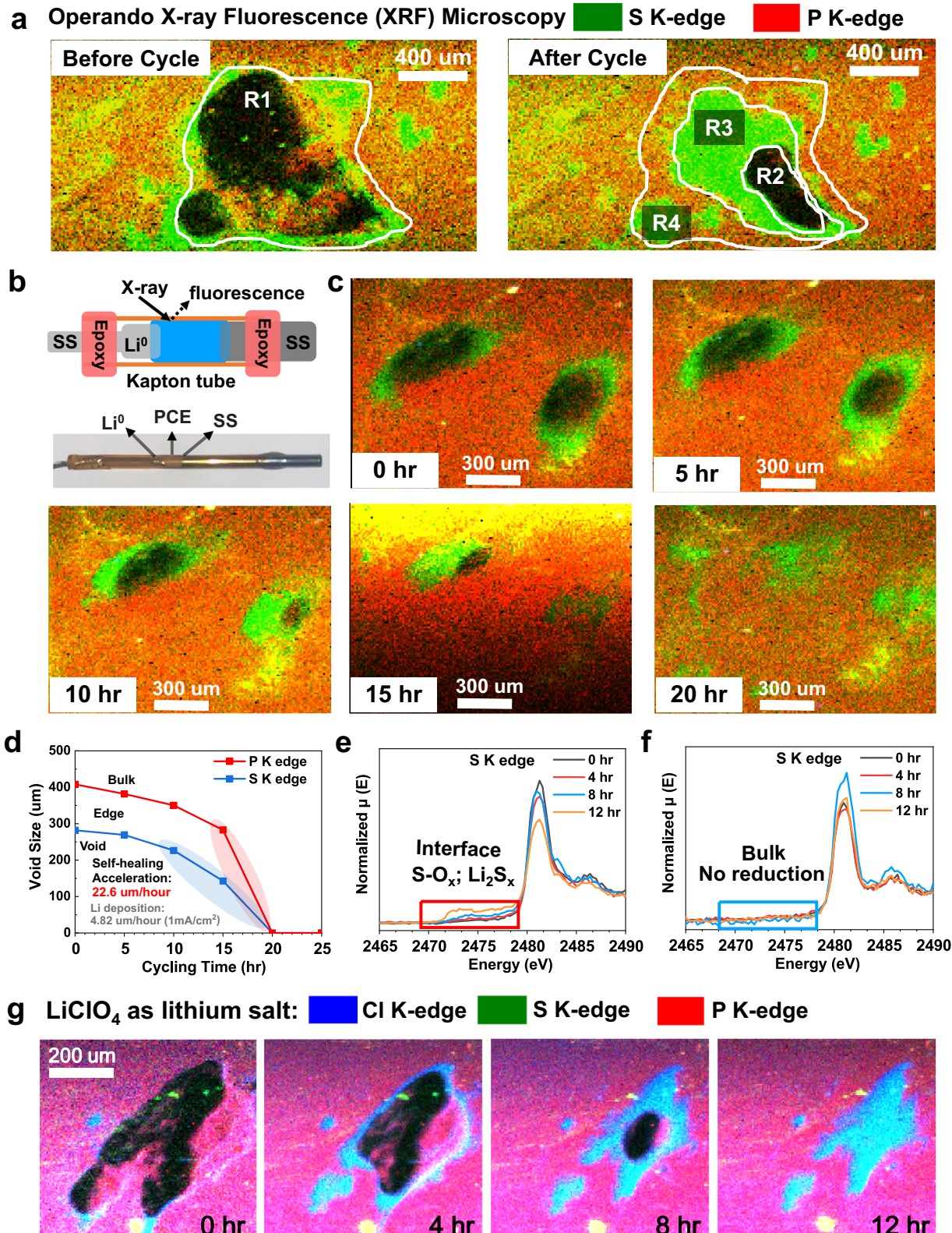

**Fig. 2 | Self-healing and SEI-forming ability revealed by *operando* synchrotronic imaging and spectroscopic techniques.** **a** Overlaid S K-edge and P K-edge XRF mappings of the PCE-based SSLMB before and after cycling at 22 °C for 12 h. **b** A schematic illustration of the tube battery and experimental set-up for XRF and XAS characterization. **c** In-situ high-resolution XRF images showing voids' self-healing dynamics at different cycling states. **d** Size evolution of the left void in XRF images showing the accelerated self-healing process. *Operando* S K-edge XAS obtained at the PCE-electrode interface (**e**) and bulk PCE area (**f**) at 22 °C. **g** Overlaid S K-edge, P K-edge, and Cl K-edge XRF images showing the individual migration trend of polymer main chain (S), lithium salt (Cl), and LATP (P).

To investigate the effect of cycling current density on the self-healing rate, we provided XRF images illustrating the self-healing of 140-$\mu$m-sized void at a cycling current density of 0.05 mA/cm$^2$ (Supplementary Figs. S29–S31). At this lower current density, the self-healing rate was 23.6 $\mu$m/hour for 142-$\mu$m voids and 50.9 $\mu$m/hour for 76-$\mu$m voids. For comparison, at 0.2 mA/cm$^2$, the self-healing rate was 22.6 $\mu$m/hour for voids sized of 226 $\mu$m (Fig. 2). These findings suggest that the self-healing rate remains relatively constant across different cycling conditions, indicating that the process is primarily diffusion-limited. During the first step of self-healing, the polymer component infiltrates the voids, creating a concentration gradient that drives the diffusion of LATP ceramics into the voids to facilitate repair. At the PCE-electrode interface (Supplementary Fig. S4), *operando* XRF results also confirmed no electrode-electrolyte delamination or crack formation during cycling, although no external pressure was applied. In the meantime, the electrode surface was gradually wetted and covered by the PCE, attributed to the excellent flexibility and self-infiltration ability of the PCE. It should be noted that the voids/cracks formation is considered the primary cause of cell failure[6,7,10]. This self-healing PCE may hold the potential for eliminating the long-existing dendrite formation and manufacturing challenges of SSEs.

*Operando* S K-edge X-ray absorption spectroscopy (XAS) was further performed to reveal the chemistry at the PCE-Li$^0$ interface. In the time-resolved XAS profiles (Fig. 2e), pristine PCE shows a major peak at 2482 eV which is ascribed to the transition of S $1s$ to $-SO_2-$ in the LiTFSI salt. With prolonged cycling, new peaks emerging between 2472 eV and 2480 eV suggests the reduction of $-SO_2-$ to lower charge-state species such as S$-O_x$[28] and $Li_2S_x$[29,30]. For comparison, XAS profiles from the bulk PCE show no noticeable change (Fig. 2f). The electrochemical reduction of LiTFSI is responsible for forming a stable SEI[31], and this is further supported by ex-situ XPS results (Supplementary Fig. S5), where S$-O_x$ and $Li_2S_x$ can be clearly identified in the S $2p$ profiles[32]. In addition, the SEI is also enriched in LiF, $Li_2O$, $Li_3N$, and $Li_2CO_3$, further suggesting the contribution of SH-SPE component to forming stable SEI. The effectiveness of this SEI in preventing LATP degradation was proved by the *operando* P K-edge XAS, where no changes in the pre-edge energy and peak shape were observed (Supplementary Fig. S6). While for the pristine LATP, the pre-peaks at 2148 eV and 2150.5 eV almost disappeared after cycling, indicating the structural change of the NASICON phase[33] due to lithiation[8] and subsequent degradation into $Li_3PO_4$ and $Li_3P$[9], which could be further supported by the XPS P $2p$ and O $1s$ profiles as shown in Supplementary Figs. S7 and S8. Finally, to confirm that the S migration originates from the dynamic polymer mainchain, we performed verification experiments by replacing LiTFSI with $LiClO_4$ (in later sections, the original PCE formulation with LiTFSI salt was employed). Consequently, the individual migration process of LATP, polymer mainchain, and lithium salt could be monitored by *operando* tracking of the P, S, and Cl elements, respectively. As shown in Fig. 2g, the polymer mainchain and $LiClO_4$ salt are found to migrate synchronously, and completely filled the 300-$\mu$m-sized void within 12 h. Based on the individual P, S, and Cl mapping in Supplementary Fig. S9, the calculated infiltration speed was 33.8 $\mu$m/hour (Supplementary Fig. S10). This result further confirms the self-infiltrating capability of the dynamically crosslinked mainchain. To the best of our knowledge, this is the first study to reveal the self-healing mechanism of a hybrid SSE, and this dynamically-crosslinking and self-infiltrating design are conceptually different from conventional hybrid SSEs which employs covalently-crosslinked or crystalized polymers.

## Structure and chemistry of deposited Li$^0$ and SEI

To demonstrate the effectiveness of this self-healing and SEI-forming PCE in enabling dendrite-free Li$^0$-anode, Fig. 3a presents representative Cryo-TEM images of the Li$^0$ deposited at 0.5 mA/cm$^2$ for 1 h (Li$^0$-Cu cell). The images were obtained in high-angle annular dark-field (HAADF)

mode (see details in Method). The results show that the deposited Li$^0$ is densely packed and has a smooth chunk morphology. Energy-dispersive spectroscopic (EDS) maps (Fig. 3b) of the deposited Li$^0$ reveal a thin uniform layer of SEI. The SEI derived from the electrochemical reduction of SPE is enriched in C, N, O, F, and S elements, which is consistent with the XAS and XPS characterization. Figure 3c presents a representative atomic-resolution Cryo-TEM image of the Li$^0$ deposit. The well-defined (110) plane (with a lattice space of 0.243 nm) of the Li$^0$ suggests that the deposited Li$^0$ has a perfect bcc structure. Figure 3d, e shows representative electron diffraction pattern (EDP) and atomic-resolution Cryo-TEM images of the Li$^0$ deposit covered by the surface SEI. It is seen that the SEI is composed of nano-sized domains (e.g., $Li_2O$ indicated by the dashed circles in Fig. 3e) with varied crystallographic orientations. Consistent with the atomic-resolution images, the Bragg spots and polycrystalline ring in the EDP can be indexed as the (110) plane of the body-centered cubic (bcc)-structured Li$^0$ and the (111) plane of the face-centered cubic (fcc)-structured $Li_2O$, respectively. From the above, the dendrite-free Li$^0$ deposition and uniform SEI morphology revealed by Cryo-TEM experiments could be ascribed to the SEI-forming and self-healing ability of PCE, and also rationalizes the excellent electrochemical properties and full cell durability, which will be discussed in the following section.

## Interphasial and interfacial ion transport

The SH-SPE also facilitate interphasial Li$^+$ ion transport by infiltrating the grain boundaries of LATP, as evidenced by cryo-TEM images in Fig. 1d. To confirm this, we conducted $^6$Li-$^6$Li 2D exchange NMR (2D-EXSY) to demonstrate Li$^+$ inter-exchange between the polymer phase and ceramic phase[34]. The two diagonal peaks in Fig. 4a correspond to self-correlation signals of SH-SPE at −2.00 ppm and LATP at −0.97 ppm, respectively. Furthermore, Li$^+$ ion exchange between SH-SPE and LATP results in off-diagonal cross-peaks, indicated by the dashed box in Fig. 4b. Increasing the mixing time ($T_{mix}$) allows more time for Li$^+$ diffusion between phases, leading to increased cross-peak intensity (Fig. 4c). Notably, the appearance of clear cross-peaks at a short mixing time of 5 ms and room temperature (Fig. 4b) indicates rapid Li$^+$ exchange between the polymer and ceramic phases[35]. In Fig. 4d, we further employed an isotope exchange method to investigate ion conduction pathways within PCE. By cycling a $^6$Li|PCE|$^6$Li cell, the $^7$Li atom in PCE is gradually replaced by the $^6$Li atom, enhancing the $^6$Li signal intensity where Li$^+$ transport occurs. As shown in the $^6$Li NMR spectra (Fig. 4e), after isotope exchange from $^7$Li to $^6$Li, the integral area for the SH-SPE signal increased by 7.1-fold, while the integral area for the LATP signal increased by 6.3-fold, which is comparable to that of SH-SPE. This result suggests that Li$^+$ ions transport equally through both the polymer and ceramic phases, due to their comparable ionic conductivities (Supplementary Fig. S36). This uniform transport mitigates the uneven deposition and dendrite formation that could arise from differences in ionic conductivity between the phases[36]. Due to the grain boundary infiltration of SH-SPE and fast interphasial ion transport kinetics, the PCE exhibits low grain boundary resistance ($R_{GB}$) of ~55 ohms ($\sigma_{GB}$ = 0.8 mS/cm) at room temperature (Fig. 4f). In comparison, pristine LATP pellet has an $R_{GB}$ and $\sigma_{GB}$ of 5000 ohms and 0.024 mS/cm, respectively. This decreased $R_{GB}$ results in high ionic conductivities of 0.75 mS/cm at 30 °C and 5.09 mS/cm at 100 °C (Fig. 4g). In addition, PCE also possesses a good Li$^+$ transference number of 0.74 (Supplementary Fig. S11), attributable to the single-ion-conducting nature of LATP ceramic.

Figure 4h shows the electrochemical impedance spectroscopy (EIS) evolution with cycling time (see equivalent circuit fitting in Supplementary Fig. S12). The Li$^0$|PCE|Li$^0$ cell showed constant and low charge transfer resistance ~14.5 ohm·cm$^2$, attributing to the stable interfacial chemistry as revealed by *operando* XAS and Cryo-TEM. In sharp contrast, Li$^0$|LATP|Li$^0$ cell showed high charge transfer resistance reaching 50000 ohm·cm$^2$ due to the uncontrolled side reactions at the

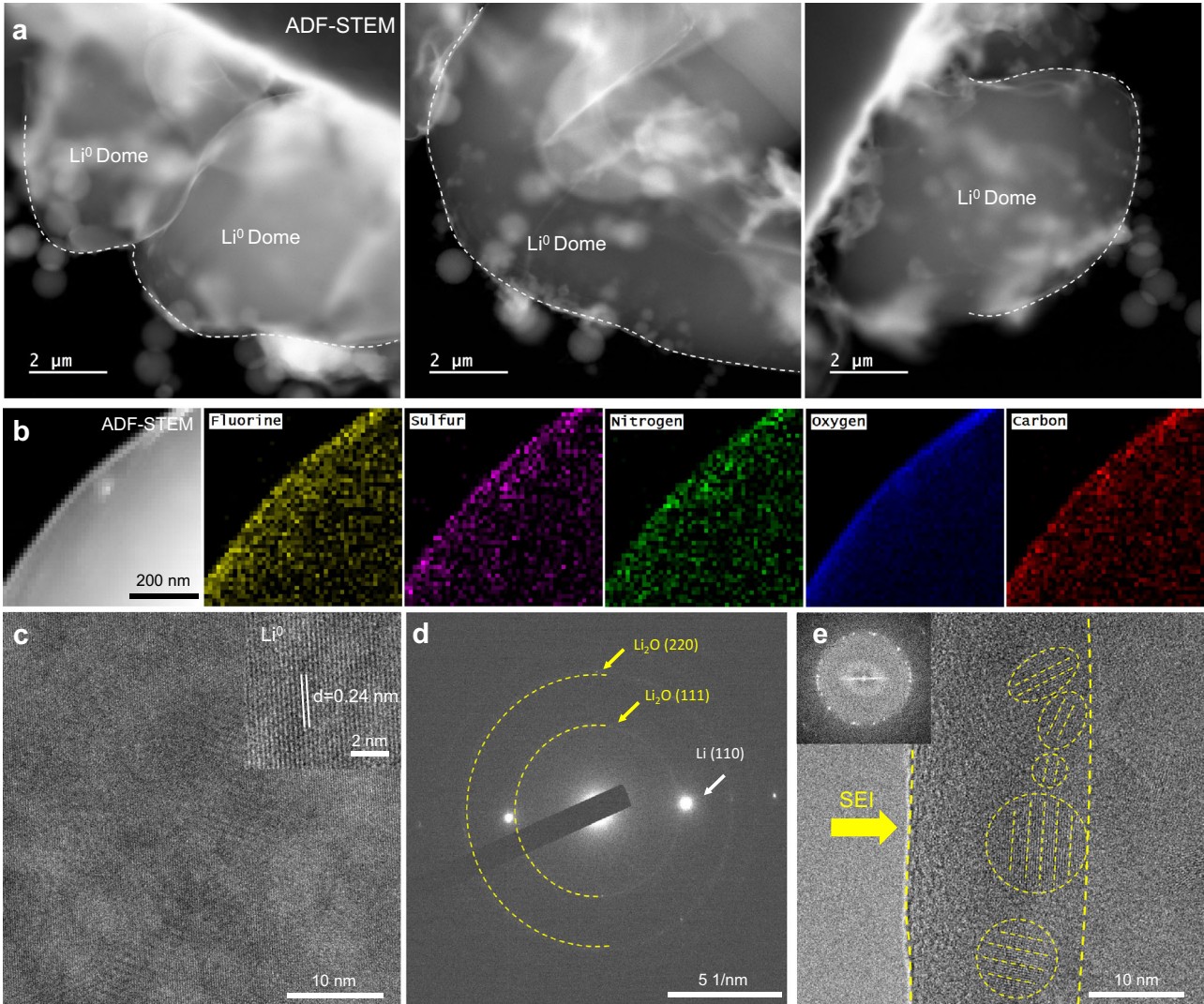

**Fig. 3 | Cryo-TEM experiments reveal densely-packed Li⁰ deposits protected by uniform SEI. a** Representative cryogenic scanning transmission electron microscopy (Cryo-TEM) images showing the smooth chunk morphology of the deposited Li⁰ with PCE. The images were obtained in high-angle annular dark-field (HAADF) mode. **b** High-angle annular dark-field STEM (HAADF-STEM) image, and energy-dispersive spectroscopic (EDS) maps of the deposited Li⁰ with the PCE electrolyte. The result shows that a thin and uniform layer of solid electrolyte interface (SEI)
enriched in C, N, O, F, and S forms on the Li⁰ surface. **c** Atomic-resolution Cryo-TEM image showing the structure of the Li⁰ deposit. The Li⁰ (110) spacing is indicated in the inset. **d** Electron diffraction pattern (EDP) of Li⁰ deposit. The Bragg spots and diffraction rings corresponding to Li⁰ and Li₂O respectively are indicated by arrows. **e** Atomic-resolution Cryo-TEM image showing the SEI on the surface of the Li⁰ deposit. The SEI comprises nano-sized domains (e.g., Li₂O indicated by dash circles) with varied crystallographic orientations.

interface[8]. SEM characterization reveals a smooth, dense, and uniform morphology of the deposited Li⁰ under PCE (Fig. 4i). Furthermore, the Li⁰|PCE|Li⁰ cell could be continuously discharged to a high areal capacity of 6 mAh/cm² (Fig. 4j), which corresponds to the plating of 29-$\mu$m-thick Li⁰. The inserted EIS plots showed constant bulk resistance, suggesting no dendrite penetration through PCE. The constant charge transfer resistance also suggests minimized side reaction and electrode-electrolyte delamination, which is hard to achieve in conventional OCE-based SSLMB[10]. Also from a manufacturing perspective, the cold-milled PCE avoids the high-temperature, high-pressure, hot-press sintering fabrication of OCEs. The SH-SPE was also synthesized via a single-step, solvent-free, and highly convenient (10 min at 22 °C) UV-polymerization approach under near-quantitative monomer conversion yield (see Methods and Supplementary Fig. S13).

## Long-term durability of PCE
Figure 5a further compares the long-term durability of Li⁰|LATP|Li⁰, Li⁰|SH-SPE|Li, Li⁰|PCE|Li⁰ cells. At a small current density of 0.05 mA/cm²,

the Li⁰|LATP|Li⁰ cell failed within 100 h, as indicated by the rapid overpotential build-up caused by interfacial side reaction (Supplementary Fig. S14). Li⁰|SH-SPE|Li⁰ cell showed improved durability at 0.2 mA|cm², then short-circuiting occurred at 1000 h (Supplementary Fig. S15). In sharp contrast, the Li⁰|PCE|Li⁰ cell stably cycled for >4000 h, and enlarged voltage-time curves (Supplementary Fig. S16) showed steady overpotential during cycling, providing further evidence that no soft-shorting occurred. This improved durability of PCE could be attributed to the combination of the ceramic with the flexible polymer filler. The polymer component can infiltrate voids and grain boundaries, eliminating 'hot-spots' for dendrite formation. Additionally, the polymer component provides more intimate interfacial contact, minimizing local current concentration and enabling stable Li⁰ deposition. It should be noted that the above performance was achieved with pristine Li⁰-anodes, i.e., without introducing any artificial SEI[37] or employing alloy-type anodes[38]. Also, all results were measured in coin cells with a stack pressure lower than 0.1 MPa. In Fig. 5b, LiFePO₄ (LFP) cathode was first paired with PCE due to its high safety,

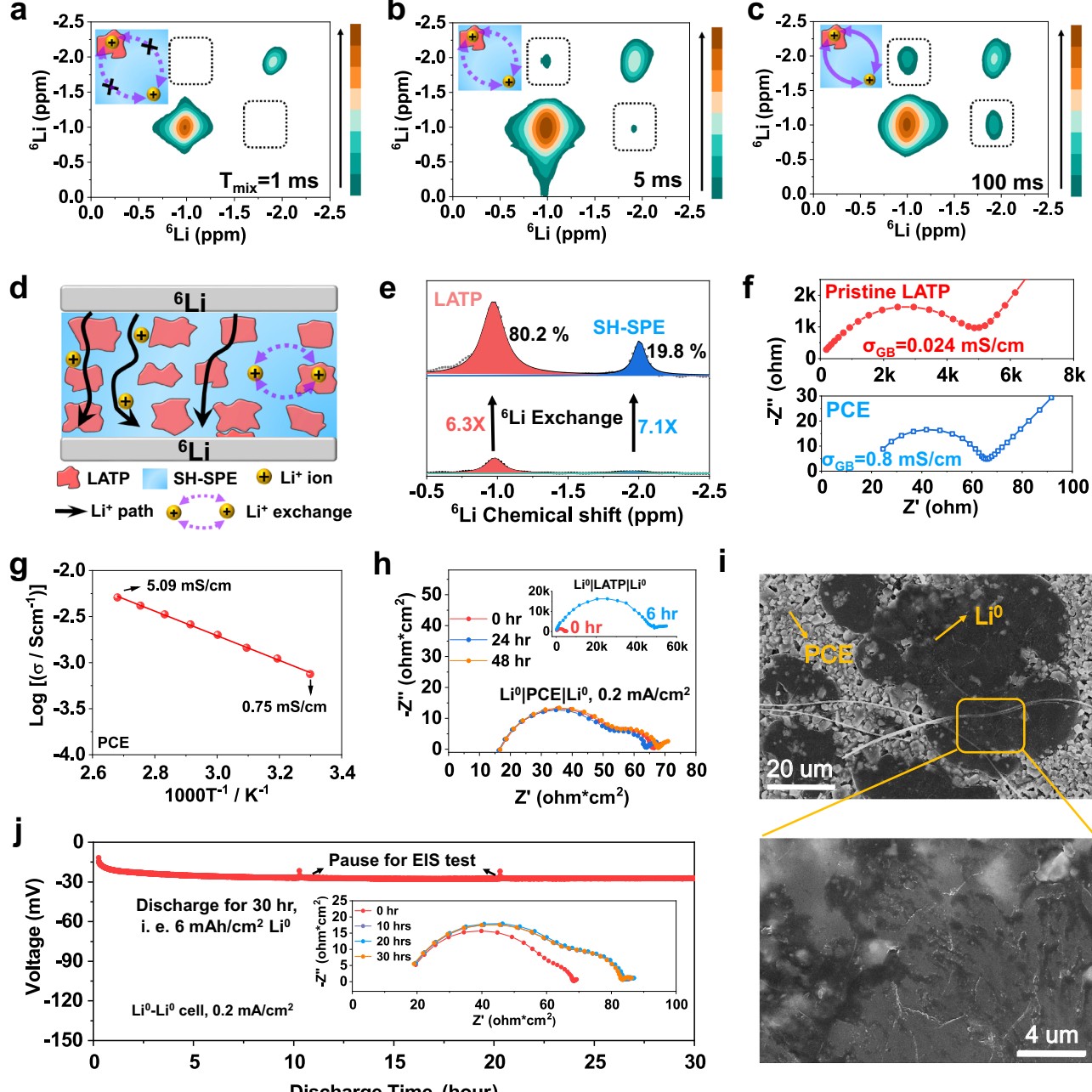

**Fig. 4 | PCE enables improved interphasial and interfacial ion transport kinetics. a–c** 2D EXSY NMR of PCE at different mixing time of 1 ms, 5 ms and 100 ms. **d** Schematic illustration of the isotope exchange method for revealing the ion conduction pathway. **e** $^6$Li solid-state NMR spectra of the pristine PCE and the PCE cycled in $^6$Li-$^6$Li symmetric cells. **f** EIS plots of SS|PCE|SS and SS|LATP|SS cells showing the grain boundary resistance at 30 °C. SS refers to stainless steel blocking electrodes. **g** Temperature-dependent ionic conductivity of PCE and the Arrhenius fitting in the form of $\sigma = Ae^{-\frac{E_a}{RT}}$. **h** Evolution of EIS plots for Li$^0$|PCE|Li$^0$ cell when cycling at 0.2 mA/cm$^2$ at 50 °C. Inset is the evolution of EIS plots for Li$^0$|LATP|Li$^0$ cell when cycling at 0.05 mA/cm$^2$ at 50 °C. **i** SEM images showing the surface morphology of Li$^0$ deposits after discharging a Li$^0$|PCE|Cu cell at 0.2 mA/cm$^2$ and 22 °C. **j** Voltage-time profile and EIS plot evolution (inserted figure) when continuously discharging a Li$^0$|PCE|Li$^0$ cell at 0.2 mA/cm$^2$ and 50 °C for 30 h.

excellent stability, and low cost. At 22 °C and C/2 (0.19 mA/cm$^2$), the Li$^0$|PCE|LFP cell showed cycling stability >1300 cycles with a high capacity retention of 92%, i.e. the capacity decay per cycle is 0.006%. The charge-discharge curves in Fig. 5c show a stable overpotential during cycling, and no sign of soft-short-circuiting could be observed. To demonstrate the compatibility with a high voltage cathode (4.3 V), we further employed a high-Ni zero-Co zero-strain cathode (LiNi$_{0.8}$Mn$_{0.13}$Ti$_{0.02}$Mg$_{0.02}$Nb$_{0.01}$Mo$_{0.02}$O$_2$, 0.6 mAh/cm$^2$) developed in our previous study[39] to pair with the PCE. The cell showed a high initial capacity of 152 mAh/g. After 500 cycles at C/2 (0.23 mA/cm$^2$) and 22 °C

(Fig. 5d, e), the retained capacity is 123 mAh/g (81%). Note that the above results were achieved under low stack pressure in coin cells (~0.1 MPa) and no catholyte was employed.

## The hierarchical architecture further improves full-cell durability

The PCE has solved the electronic conductivity, grain boundary resistance, mechanical instability, and fabrication challenges of OCEs, thus dramatically improving the durability of LATP from <100 h at 0.05 mA/cm$^2$ to ~2000 h at 1 mA/cm$^2$. However, considering the intrinsic poor

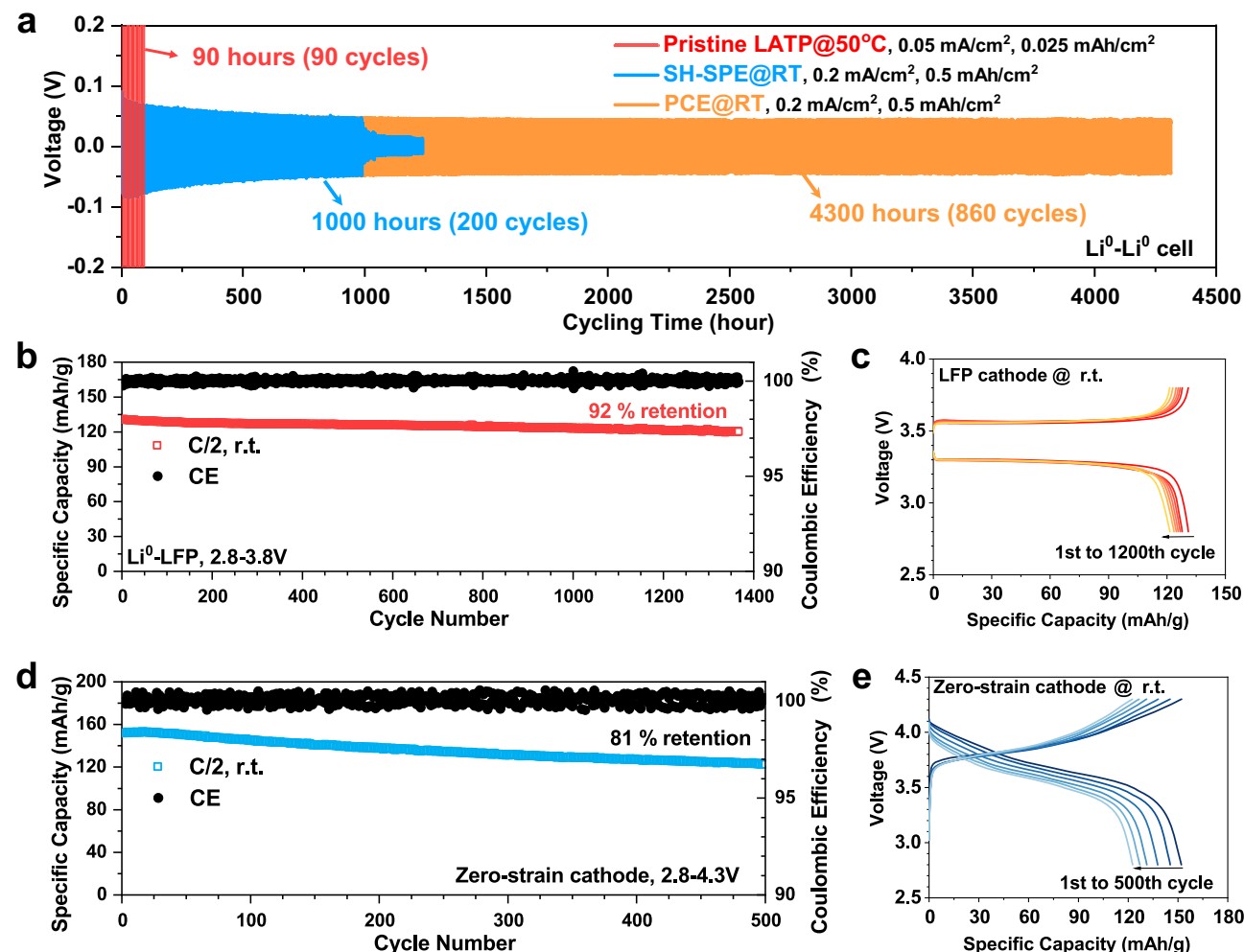

**Fig. 5 | Long-term durability of PCE. a** Durability comparison of LATP (50 °C), SH-SPE (22 °C), and PCE under Li⁰-Li⁰ cell configuration (22 °C). Cycling stability (**b**) and charge-discharge curves (**c**) of Li⁰|PCE|LFP cells at 22 °C and C/2. The loading of the LFP cathode is 2.4 mg/cm². Cycling stability (**d**) and charge-discharge curves (**e**) of PCE-based full cell employing the zero-strain cathode at 22 °C. The loading of zero strain cathode is 2.45 mg/cm².

electrochemical stability of LATP against Li⁰-anode and the thin coating of SH-SPE on LATP, side reactions are still possible after long-term cycling. Therefore, we further developed a hierarchical architecture (H-SSE, Fig. 6a) comprising a polyacrylate-based SPE (PA-SPE, 120-μm-thick) as the buffer layer, and the PCE as the dendrite-inhibiting layer (see Methods for details about PA-SPE). XPS profiles in Fig. 6b and Supplementary Fig. S17 show that no Ti, Al, or P was detected in the H-SSE-derived SEI. Instead, the SEI is mainly composed of LiF, Li₃N, Li₂O, Li₂CO₃, and SOₓ originating from the electrochemical reduction of PA-SPE, suggesting the complete isolation of LATP from Li⁰ anode. In addition to optimizing the interfacial chemistry, the PA-SPE buffer layer also enables an extremely low electron conductivity of 2.3E-9 S/cm at 22 °C (Fig. 6c), which is 652-fold lower than the pristine LATP. Supplementary Figs. S18–S27 summarizes the additional electrochemical properties of H-SSE. H-SSE shows a high ionic conductivity of 1.01 mS/cm at 30 °C (Supplementary Fig. S18a) and a wide electrochemical stability window of 0–4.6 V vs. Li⁺/Li (Supplementary Fig. S18b). Smooth and dendrite-free morphology of plated Li⁰ was revealed by both SEM (Supplementary Fig. S19) and cryo-TEM characterization (Supplementary Fig. S20). Under Li⁰|H-SSE|Li⁰ cells, high critical current density (>30 mA/cm², Supplementary Figs. S21, 22) and further improved durability were achieved. The Li⁰|PCE|Li⁰ cell was then cycled at current density of 1 mA/cm², demonstrating stable cycling for 2900 h (Supplementary Fig. S23). The total amount of plated Li⁰ during cycling can be quantified by the accumulated areal

capacity (AAC), calculated as AAC = Current density (mA/cm²) × Cycling time (hours). Consequently, when cycling at 1 mA/cm², the AAC of the Li⁰|PCE|Li⁰ cell reached 2900 mAh/cm². When cycling the cell at a higher current density of 2 mA/cm² and a large cut-off areal capacity of 2 mAh/cm², the cell also delivered an AAC of 2000 mAh/cm² (Supplementary Fig. S24). In addition, we further cycled the Li⁰|PCE|Li⁰ cells at higher current densities of 5 mA/cm², 10 mA/cm², and 20 mA/cm², with cut-off areal capacities of 2 mAh/cm², 1 mAh/cm², and 0.5 mAh/cm², respectively. As shown in Supplementary Figs. S25–S27, the AAC reached 1400 mAh/cm² and 1500 mAh/cm² when the cells were cycled at 20 mA/cm² and 10 mA/cm², respectively. To confirm this excellent dendrite inhibiting capability originates from the PCE, we also provide the cycling stability of the PA-SPE buffer layer (without PCE) in Supplementary Figs. S23, 24. Li⁰|PA-SPE|Li⁰ cells exhibit a much shorter cycling life of <200 h at 1 mA/cm² and <50 h at 2 mA/cm², due to the absence of ceramic components to block the dendrites.

Next, the H-SSE was paired with different cathodes. Li⁰|H-SSE|LFP cell showed good rate capability (Fig. 6d) by delivering a specific capacity of 140 mAh/g at C/2 (0.16 mA/cm²) and 114 mAh/g at 3 C (0.99 mA/cm²). At 50 °C and 2 C (0.69 mA/cm²), the LFP-based cells showed cycling durability of ~4000 cycles with a high capacity retention of 88%, i.e., the capacity fade per cycle is 0.003% (Fig. 6e). After cycling at C/2 (0.38 mA/cm²) and 22 °C for 2400 cycles (loading = 4.5 mg/cm²), the capacity retention is 84%. Due to its wide electrochemical window and excellent interface with Li⁰-anode, the H-SSE is also

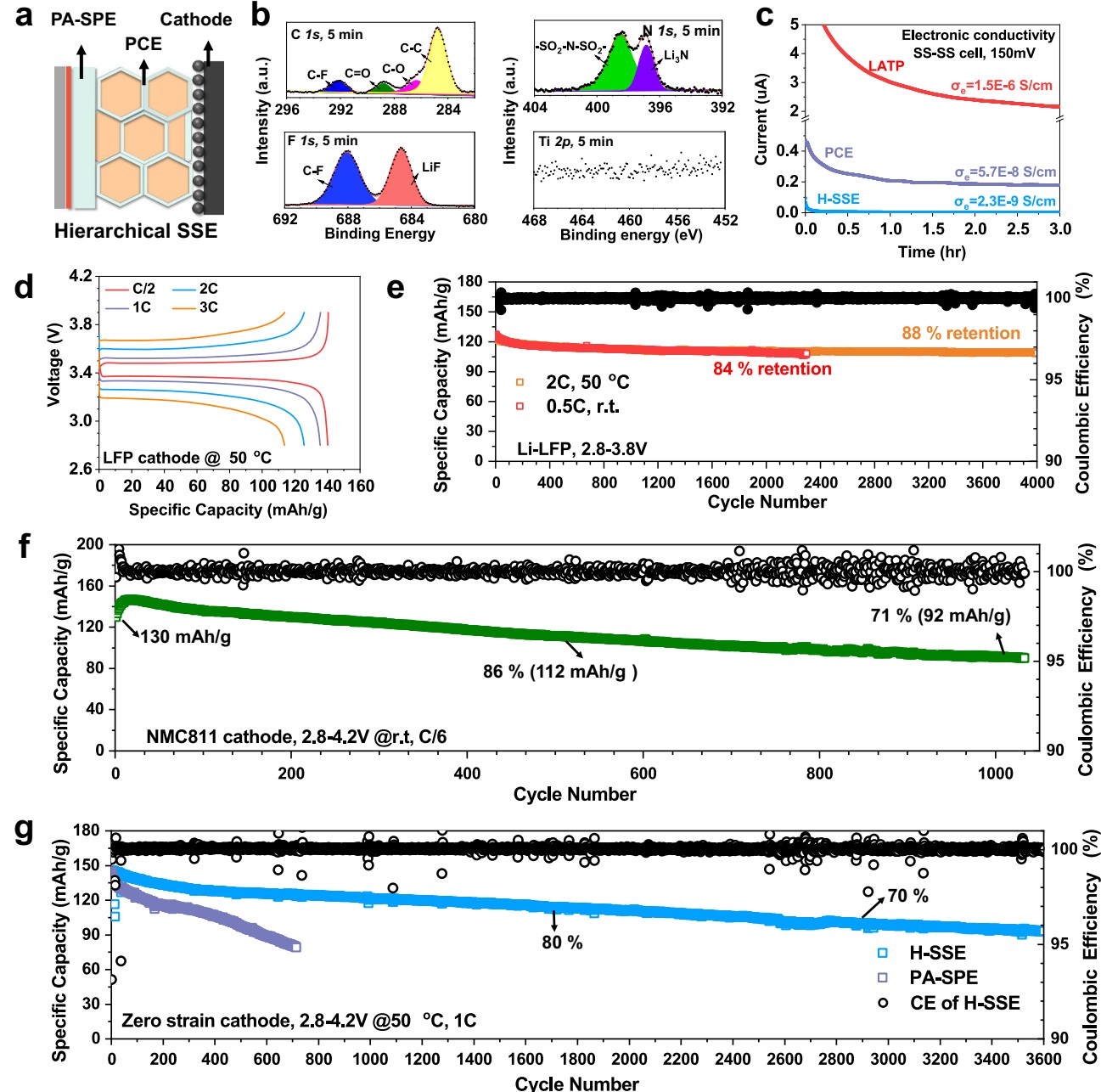

**Fig. 6 | Hierarchical SSE (H-SSE) with a PA-SPE buffer further extends the cycling durability by eliminating LATP degradation. a** Schematic illustration showing the architecture of H-SSE. **b** C *1s*, F *1s*, N *1s*, and Ti *2p* XPS profiles of the H-SSE derived SEI after Ar sputtering for 5 min. **c** Electron conductivities at 22 °C of LATP, PCE, and H-SSE measured under an SS-SS cell configuration, 150 mV polarization, and r.t. conditions. **d** Voltage-capacity curves of Li$^0$|H-SSE|LFP full cell at 50 °C showing the rate capability. The loading of LFP is 1.95 mg/cm$^2$. **e** Performance of H-SSE-based full cells when employing LFP at 22 °C (4.48 mg/cm$^2$) and 50 °C (1.95 mg/cm$^2$). **f** Performance of H-SSE-based full cells at 22 °C when employing high-loading NMC811 (7.4 mg/cm$^2$). **g** Performances of H-SSE-based and PA-SPE-based full cells at 50 °C when employing a high-Ni, zero-Co, zero-strain cathode (2.89 mg/cm$^2$). The right y-axis displays the Coulombic efficiency of the H-SSE-based full cell, while the Coulombic efficiency of the PA-SPE-based full cell is shown in Supplementary Fig. S32.

compatible with a commercial high-loading LiNi$_{0.8}$Mn$_{0.1}$Co$_{0.1}$O$_2$ (NMC811) cathode. Due to its high loading (7.4 mg/cm$^2$), the r.t. performance of the Li$^0$-NMC811 cell was evaluated at a lower C-rate of C/6 (0.24 mA/cm$^2$). As shown in Fig. 6f, the cell showed an initial capacity of 129.9 mAh/g, and the capacity after 640 and 1000 cycles was 104 and 92 mAh/g (80% and 71% retention). The charge-discharge curves depicted in Supplementary Fig. S28 further prove no soft-shorting occurred despite the large cycling capacity. Note that the cell was cycled at low stack pressure in coin cells (0.1 MPa), room temperature (21 °C), and without any catholyte.

To further improve the cycling stability, the 4.2 V high-Ni zero-Co zero-strain cathode (LiNi$_{0.8}$Mn$_{0.13}$Ti$_{0.02}$Mg$_{0.02}$Nb$_{0.01}$Mo$_{0.02}$O$_2$, 0.6 mAh/cm$^2$)[39] was paired with the H-SSE (Fig. 6g). A high initial capacity of 144 mAh/g was achieved at 1 C (0.55 mA/cm$^2$) and 50 °C. The cell then cycled stably for 3600 cycles (6 months) without short-circuiting. The capacity retentions at 1650 and 2860 cycles were 80% and 70%, respectively. For comparison, the capacity of PA-SPE-based cells (without PCE) declined from 144 to 80.5 mAh/g within 700 cycles. The capacity fade rate of 0.063% per cycle is 6-fold higher than the H-SSE-based cell (0.010%), indicating the critical role of PCE as a dendrite

inhibiting layer to improve full cell durability. To further evaluate the potential of the PCE in practical applications, we tested the performance of H-SSE when paired with the zero-strain cathode in pouch cells (Supplementary Fig. S37). Despite the absence of external pressure, the pouch cell demonstrated an initial discharge capacity of 141.8 mAh/g at 1 C (0.58 mA/cm$^2$) and 50 °C, maintaining good durability over 400 cycles. The capacity retention was 79% after 400 cycles, and the average coulombic efficiency from the 1 st to the 400th cycle was 99.94%. In Supplementary Tables S1–S3, we further compared battery performance with previously reported solid polymer electrolytes. The PCE exhibits notable improvements in Li$^0$ anode compatibility, durability, and areal capacity, thereby demonstrating the effectiveness of our design in achieving durable and high-performance solid-state batteries.

In summary, we have developed a cold-milled plastic ceramic electrolyte (PCE) that avoids the high-pressure, high-temperature, hot-press sintering fabrication of conventional OCEs. The self-healing capability originated from the aprotic and dynamically crosslinked polymer network was demonstrated by *operando* XRF characterization, and is crucial for eliminating the defects/cracks/voids of SSE and maintaining its mechanical integrity. Infiltrating the grain boundaries with the SH-SPE has delivered dramatically enhanced ionic conductivity and decreased electron conductivity. Completely isolating the LATP from Li$^0$-anode via the dual protection from the PA-SPE buffer layer and SH-SPE grain boundaries protection layer has enabled robust SEI and full cell durability >4000 cycles. Despite demonstrating good cycling durability, the initial low coulombic efficiency and discharge capacity need further improvement. These issues may stem from potential side reactions at the cathode-electrolyte interface and high resistance at room temperature, which contributes to voltage hysteresis[40]. Future studies will focus on enhancing the initial coulombic efficiency by developing a stable cathode-electrolyte interface (CEI)[41] and addressing the low discharge capacity through the design of composite cathodes to improve ion conduction within the electrode. Overall, this study has addressed the conductivity, interface, mechanical, stack pressure, and fabrication challenges of solid-state Li$^0$-anode batteries and demonstrated long cycle life, high current density, and high areal capacity full cells. This study also provides approaches for the SSE community: to solve the electrochemical/mechanical failures of inorganic electrolytes via combing polymer electrolytes with functionality like SEI-forming, self-healing, and stimuli-responsiveness.

## Methods
### Materials
Ethyl acrylate (EA, 99%), ethylene glycol dimethylacrylate (EDA, 98%), azobisisobutyronitrile (AIBN, 98%), lithium perchlorate (LiClO$_4$, 99.99%), and 4-fluoro-1,3-dioxolan-2-one (FEC, 99%) were purchased from Sigma Aldrich and used as received. Phenylbis(2,4,6-trimethylbenzoyl) phosphineoxide (PPO, 96%), lithium bis (trifluoromethanesulfonyl) imide (LiTFSI, 98%), and succinonitrile (SN, 99%) were purchased from TCI. (Trifluoromethane) sulfonimide lithium methacrylate (MTFSI) was purchased from Specific Polymer and used as received. Li$_{1.5}$Al$_{0.5}$Ti$_{1.5}$(PO$_4$)$_3$ (LATP, 99.9%) with particle size of 1–5 μm were purchased from MSE Supplies and used as received.

### Synthesis of the self-healing solid polymer electrolyte (SH-SPE)
The SH-SPE was synthesized via a solvent-free, one-pot UV-polymerization method[42]. EA and LiMTFSI were employed as the monomers because the non-covalent bonding between –CH$_3$ and –CF$_3$ groups could provide dynamic and revisable interaction between polymer chains to enable self-healing function[24]. SN was employed as a solid crystal plasticizer due to its non-flammability, solid nature, and excellent solvating ability to lithium salt[43]. FEC (5 wt%) was used as the SEI forming additive due to its well-document capability to form a LiF-

rich SEI[44]. Note that no covalent crosslinker was employed to ensure the efficient infiltration of polymer into the voids/cracks. Experimentally, EA (0.35 g), LiMTFSI (0.35 g), SN (1 g), LiTFSI (0.6 g), and FEC (0.11 g, 5 wt%) were mixed without the addition of any solvent and stirred for 3 h to form a homogenous precursor. PPO (0.1 wt%) as the photoinitiator was then added to the precursor. After stirring for 5 mins, the precursor was poured onto a clean glass plated and exposed to UV irradiation (365 nm) for 10 mins. The obtained SH-SPE was then stored in the glove box before use.

The monomer conversion yield and the chemical structure of SH-SPE were determined by $^1$H NMR spectra (Supplementary Fig. S13). As shown in the spectra of EA and MTFSI, signals 1–2 and 9–11 at 5.5–6.2 ppm were assigned to protons on the C=C double bonds. After UV polymerization, these signals completely disappeared, and the emerging signals 18–19 at 1.5–2.5 ppm were assigned to the polymer backbone of SH-SPE. This result suggests a near quantitative (-100%) monomer conversion yield after UV polymerization. In the spectra of SH-SPE, signal 13 at 1.1 ppm and signal 16 at 4 ppm were assigned to the polymerized EA and MTFSI units, respectively, suggesting the successful introduction of both the proton-donating group and the proton-accepting group into the SH-SPE.

### Synthesis of the plastic ceramic electrolyte (PCE)
Inside an Ar-filled gloved box, the SH-SPE (1.5 g) and LATP (3.5 g) were hand-milled in an agate mortar at 22 °C. Afterwards, the obtained PCE was roll-pressed to a thin film, folded, and roll-pressed again. The above process was repeated multiple times until a homogenous solid membrane was obtained. The PCE was then stored inside the glove box overnight to ensure the well-infiltration of SH-SPE into the grain boundaries of LATP. The thickness of PCE was 350 μm (Supplementary Fig. S3).

The incorporation of 30 wt% SH-SPE aims to reduce grain boundary resistance, impart self-healing capabilities, form a stable SEI, and simplify the electrolyte fabrication process. By decreasing grain boundary resistance and enhancing ionic conductivity, the PCE demonstrates improved battery performance at room temperature. The flexibility and self-healing properties of the PCE reduce mechanical failure and dendrite penetration during battery operation, thereby mitigating safety hazards related to short-circuiting. Additionally, the cold-milling process and improved processability enable the fabrication of thinner PCE layers compared to conventional ceramic electrolytes, leading to an increased energy density.

The modulus of SH-SPE and PCE was measured using a rheometer. Supplementary Fig. S34 shows the viscoelasticity of SH-SPE, indicating the change in storage modulus (G′) and loss modulus (G″) with increasing strain rate. SH-SPE exhibits a G′ range from 10$^4$ to 10$^5$ Pa. As the strain rate changes, G′ consistently exceeds G″, confirming the solid nature of SH-SPE[45]. With the introduction of 70 wt% LATP ceramic, G′ dramatically increases to 10$^6$-10$^7$ Pa, which could be due to particle-particle friction enhancing the mechanical rigidity of PCE[46,47]. Supplementary Fig. S35 also illustrates the stretchability of SH-SPE and PCE, both of which can be stretched to over 220% of their original length without breaking. This suggests good flexibility, which is ideal for avoiding mechanical fractures during battery fabrication and operation.

Compared with traditional polymer electrolytes, the PCE combines the advantages of both polymer and ceramic electrolytes. The high modulus of ceramic particles provides a physical barrier to block dendrite penetration, thereby extending the cycling life of Li$^0$-anode batteries. Additionally, the single Li$^+$ conducting nature of LATP ceramic enables a high Li$^+$ transfer number (t$_{Li+}$) of 0.74 (Supplementary Fig. S11), which enhances effective Li$^+$ conductivity and minimizes dendrite formation by reducing concentration polarization[48]. More importantly, this PCE demonstrates a uncommon dual-phase self-healing mechanism with a fast healing rate of 22.6 μm/hour, effectively eliminating "hot spots" for dendrite formation, such as voids and cracks.

Compared with previous self-healing solid electrolytes, this study employed aprotic $-CH_2-CH_3$ moieties, avoiding the use of $Li^0$-reactive $-OH$ and $-NH$ moieties[26,27], which enables better interfacial stability with the $Li^0$ anode. Additionally, this study reveals and quantifies the real-time self-healing mechanism during battery cycling, demonstrating a fast self-healing rate of 22.6 μm/h. Furthermore, the PCE exhibits a dual-phase self-healing process, allowing ceramic particles to migrate through the polymer network and fill voids. The high modulus of the ceramic particles could also contributes to dendrite inhibition. These advantages have led to significantly enhanced cycling durability compared with previous self-healing solid electrolytes (Supplementary Table S3). In practical applications, batteries might be charged once every 24 h and discharged over a period of 10–50 h. These typical usage patterns and charging cycles align well with the self-healing rate of our PCE.

## Synthesis of the PA-SPE buffer layer

The PA-SPE was synthesized via a thermal polymerization pathway developed in our previous study[49]. Similar to SH-SPE, SN was employed as a solid crystal plasticizer due to its non-flammability, solid nature, and excellent solvating ability to lithium salt[43]. FEC was used as the SEI forming additive due to its well-document capability to form a LiF-rich SEI[44]. Experimentally, the EA (0.3 g), EDA (0.3 g), SN (1 g), LiTFSI (0.6 g), and FEC (0.11 g, 5 wt %) were mixed without the addition of any solvent and stirred for 1 h to form a homogenous liquid precursor. AIBN (0.1 wt %) as the thermal initiator was then added to the precursor. After stirring for 5 min, the liquid precursor was poured onto a glass fiber reinforcement and sandwiched between two pieces of stainless steel. After heating at 65 °C overnight to initiate the polymerization of monomers, the obtained PA-SPE was peeled-off from the stainless steel and stored in the glove box before use. The thickness of PA-SPE was ~120 μm.

## Electrode preparation and battery assembly

The LFP and the high-Ni, zero-Co, zero-strain cathode[50] were prepared by a slurry-casting method. Active material (80 mg), superP (10 mg), and PVDF binder (10 mg in an 8 wt% NMP solution) were mixed with a Thinky Mixer® and then cast onto an Al foil. After drying under vacuum at 80 °C for 20 h, the obtained electrode was punched into 12 mm discs and stored in the glove box before use. The high-loading NMC811 cathode was provided by the CAMP Facility at Argonne National Laboratory. $Li^0$ foil with diameter of 12 mm and thickness of 450 μm or 250 μm was employed as the anode. Solid-state batteries were assembled under a 2032-type coin cell configuration inside the glove box with water content <1 ppm and $O_2$ content <0.1 ppm. The obtained batteries were tested on a NEWARE multichannel cycler. To confirm electrolyte infiltration and mixing with the cathode, SEM-EDS mapping of the cross-section of an NMC811 cathode after cycling with H-SSE is provided in Supplementary Fig. S33. N and S elements from the polymer component were found to be homogeneously distributed within the cathode. This suggests that lithium salt (LiTFSI) and succinonitrile solid crystal plasticizer could infiltrate the pores of the cathode layer, creating an ion conduction pathway. As a result, H-SSE can cycle with commercial pre-cast cathodes without the need for additional ceramic or polymeric ionic conductors, enhancing the compatibility of our electrolyte with conventional battery fabrication techniques.

## Electrochemical characterization

Electrochemical impedance spectroscopy (EIS) profiles of $Li^0$|SSE|$Li^0$ or SS|SSE|SS cells were obtained under a frequency range from 1 MHz to 1 Hz and polarization voltage of 5 mV. Ionic conductivities of the SSEs were calculated using Eq. (1):

$$\sigma_t = d/(R_t \cdot S) \tag{1}$$

where $R_t$ (ohm), S (cm$^2$), and d (cm) are the resistance, area, and thickness of the prepared SSEs in the SS|SSE|SS cells.

Cyclic voltammetry (CV) was employed to measure the electrochemical stability window (ESW) of prepared SSEs. $Li^0$|SSE|SS cells with $Li^0$ as the reference electrode and SS as the blocking electrode was firstly assembled and then scanned at 1 mV/s from open circuit potential to 4.6 V and then with step-down scanning voltages from 4.6 V to −0.3 V.

$Li^+$ transference number ($t_{Li+}$) was measured under the cell configuration of $Li^0$|SSE|$Li^0$ employing the potentialstatic polarization method established by Bruce and Vincent[51]. The $t_{Li+}$ was calculated using Eq. (2):

$$t_+ = \frac{I_{ss}(\Delta V - I_0 R_0)}{I_0(\Delta V - I_{ss} R_{ss})} \tag{2}$$

Where $\Delta V$, $I_0$, and $I_{ss}$ are the polarization voltage, the current at the initial state, and the current at the steady state, respectively. $R_0$ and $R_{ss}$ are the charge transfer resistance at the initial and steady-state, respectively. The measurements of ionic conductivity and transference number were performed once for each sample.

## Material characterization

SEM characterization was conducted using a LEXI-FEI Magellan400. The $Li^0$ deposits for SEM characterization was obtained after discharging a $Li^0$|PCE|Cu cell or $Li^0$|H-SSE|Cu at 0.2 mA/cm$^2$ at 22 °C. Kratos AXIS-Supra was employed to record the X-ray photoelectron spectroscopy profiles. To characterize the SEI, $Li^0$-$Li^0$ symmetric cells employing different electrolytes were cycled at 0.2 mA/cm$^2$ at 22 °C. Following cycling, the coin cells were disassembled to retrieve the cycled $Li^0$ anode. All samples for XPS measurement were transferred under an inert atmosphere through an Ar-filled glove box. The viscoelastic properties of SH-SPE and PCE were evaluated using a TA DHR-2 rheometer in oscillation mode with a parallel plate configuration. Solid-state NMR experiments were carried out utilizing a Bruker Avance 500 spectrometer operating at a magnetic field strength ($B_0$) of 11.7 T. All experiments were conducted using a Bruker double-resonance MAS probe. The Larmor frequencies for $^7Li$, $^6Li$ and $^1H$ nuclei were 194.37 MHz, 73.6 MHz, and 500.13 MHz, respectively. Calibration of the spectrometer for $^7Li$ and $^6Li$ experiments was performed relative to a 1 M LiCl solution (set at 0 ppm). Calibration of the spectrometer for $^1H$ experiments was performed relative to tetramethylsilane (TMS) (set at 0 ppm). Samples were packed into a 4 mm diameter $ZrO_2$ rotor, with a spinning rate set at 8000 Hz. To analyze the ion conduction pathway, $^6Li$ isotope exchange experiments were conducted. The PCE electrolyte underwent cycling in a $^6Li$-$^6Li$ symmetric cell for 160 h to replace $^7Li^+$ ions with $^6Li^+$ ions. Subsequently, the cell was disassembled, and the PCE electrolyte was loaded into the $ZrO_2$ NMR rotor within a glove box. The signal intensity of the resulting $^6Li$ spectrum was normalized based on sample mass and the number of scans, and then compared with that of the pristine PCE.

## Tender energy X-ray fluorescence (XRF) microscopy and XAS characterization

XRF microscopy and XAS spectra were obtained at beamline 8-BM (TES) of the National Synchrotron Light Source II (NSLS II) at Brookhaven National Laboratory. A tube battery geometric with $Li^0$ as the reference electrode, PCE as the electrolyte, and stainless steel (SS) as the working electrode was assembled inside a Kapton® tube and sealed with epoxy resin to avoid air exposure. A detailed procedure for assembling the in-situ cells is described as follows: lithium metal was melted at 200 °C inside the glove box. Stainless steel wire (SS, 0.8-mm-diameter) was then dip-coated by the liquid-state lithium metal. The prepared $Li^0$-anode was inserted into a 30-mm-length, 2-mm-diameter Kapton® tube, and then sealed with epoxy resin. PCE and SS rod (2-mm-diameter) were then inserted through another side of the tube and the tube end was then sealed with epoxy resin (see Fig. 2b and Supplementary Fig. S38). In this

experiment, the thickness of PCE is around 2 mm to provide a large volume size for better observing the self-healing process. In other electrochemical measurements, the thickness of PCE was ~350 μm. The as-obtained in-situ cell was perfectly sealed in an aluminum-coated plastic bag before use. For XRF and XAS experiments, the cell was quickly transferred to a He-filled chamber. The chamber was continuously purged by He to maintain $O_2$ content <0.1%. For XRF data analysis, the area of the void was measured by ImageJ® software. The size of the void was defined as the square root of the void area.

## Cryo-TEM experiment

For the cryo-TEM experiments, a Gatan single-tilt liquid nitrogen holder was used to transfer the samples under frost-free conditions at −196 °C (liquid nitrogen). The TEM experiment was performed on JEOL transmission electron microscopes operated at 200 and 300 KeV (JEOL 2100F and JEOL GrandArm). For the Cryo-TEM sample preparation, the TEM grid was placed on a Cu disc and then employed as the working electrode in coin cells. After $Li^0$ deposition at 0.5 mA/cm² for 1 h, the cell was disassembled, and the TEM grid was sealed in an aluminum-coated plastic bag. The TEM grid was then quickly plunged into a liquid nitrogen bath, loaded onto the precooled cryo holder, and finally transferred into the TEM.

## Data availability

The data generated in this study are provided within in the Supplementary Information/Source Data file. Source data are provided with this paper. Additional data related to this research can be obtained from the corresponding authors upon request. Source data are provided with this paper.

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

## Acknowledgements

This work is primarily supported by the Assistant Secretary for Energy Efficiency and Renewable Energy, Vehicle Technology Office of the U.S. Department of Energy (DOE) through the Advanced Battery Materials Research Program under contract no. DE-SC0012704 (H.L.X., E.H.). The Cryo-TEM characterization was supported by the Office of Basic Energy Sciences of the U.S. Department of Energy, under award no. DE-SC0021204 (H.L.X.). This research used beamline 8-BM of the National Synchrotron Light Source II, a U.S. Department of Energy (DOE) Office of Science User Facility operated for the DOE Office of Science by Brookhaven National Laboratory under Contract No. DE-SC0012704. This work made use of facilities and instrumentation at the UC Irvine Materials Research Institute (IMRI), which is supported in part by the National Science Foundation through the UC Irvine Materials Research Science and Engineering Center (DMR-2011967). The NMR experiment was performed by Dr. John Kelly at the NMR facility at the UCI Department of Chemistry. XPS work was performed using instrumentation funded in part by the National Science Foundation Major Research Instrumentation Program under grant no. CHE-1338173. This research used resources from the Center for Functional Nanomaterials (CFN), which is a U.S. Department of Energy Office of Science User Facility, at Brookhaven National Laboratory under Contract No. DE-SC0012704. We would like to acknowledge the ANL CAMP facility for providing the electrodes. The CAMP Facility is fully supported by the DOE Vehicle Technologies Office.

## Author contributions

H.L.X. conceived the idea. Y.H. designed the experiment. Y.H. performed the electrolytes and electrodes' design, synthesis, and performance tests. C.W. performed the cryo-EM experiments and data interpretation. S.B., R. L., Y.H. and Y.D. conducted the XRF and XAS experiments. R.Z. conducted the SEM characterization. P.Z. conducted the XPS studies. Z.C. contributed to the electrolyte synthesis. S.E.T. provided the NMC811 cathode. E.H. contributed to the writing and editing of the manuscript. Y.H., C.W. and H.L.X. wrote the manuscript with inputs from all authors.

## Competing interests

The authors declare no competing interests.
