## [Transparent Peer Review file · Nature Communications]

A self-healing plastic ceramic electrolyte by an aprotic dynamic polymer network for lithium metal batteries

Corresponding Author: Professor Huolin Xin

Version 0:

Reviewer comments:

Reviewer #1

(Remarks to the Author)

This is a comprehensive and well presented set of work focused on synthesis of novel self healing polymers for ceramic Li⁺ conducting electrolytes and includes a high level of characterisation and analysis. The cycling and lifetime performance achieved through these material innovations is impressive and the conclusions drawn are largely valid from the data presented. I suspect the work will be impactful in the field if published.

I have returned a word copy of the document including tracked changes for minor corrections suggestions, and comments where I have questions or require more clarity. These comments are also summarised as below:

- 1) Figure 1: It's not clear what the lighter and darker materials are in these photos, and if the 'self healed' has formed from the crack in the top picture, and if so, how. Some labelling and increased info in the caption would be useful here
- 2) What is the driving force for migration of LAMP particles to fill the voids? Why would this happen?
- 3) Does the speed of self healing (20 hours, page 4) depend on the cycling or plating/stripping rate? Or is it more like a diffusion limited process?
- 4) The improved durability (page 11) is attributed to high modulus of ceramic components. But pure OCEs have high modulus and still suffer dendrites and cracking - how is this explained? Perhaps it is not the high modulus of the ceramic component, but the combination of the ceramic with the flexible polymer 'filler' in between that does not provide sites for Li plating and therefore dendrites/shorts through the electrolyte
- 5) Figure 5 a - How many cycles does the time correspond to? Can this be added to the figure as a label or in the caption?
- 6) Page 13 some 'cycling' mentions only an accumulated capacity and some mentions a cycle capacity, which leads me to think that some of this cycling is 'charge discharge' to some set capacity limit, and some is 'one way' to a total plating capacity. But it's not clear why, if so, there are different protocols. If so, clarify if this is different to the 1 mA/cm² and 2900 mAh/cm² AAC in the previous experiment
- 7) Page 13 - Why is the rate used for NMC811 so much lower than for LFP? The initial capacity is also lower than expected for 811. Could the authors note and explain this in the text?
- 8) Figure 6g - Is the efficiency in this figure referring to the H-SSE or PA-SPE? Specify, and add in the other if missing. Change 'Efficiency' to Coulombic Efficiency' on axis labels for these graphs
- 9) The high loading NMC cathode - Was this provided as pre-cast electrodes, and if so, how did you ensure electrolyte infiltration/mixing with the cathode? Was a composite cathode/SSE made?
- 10) Figure S4 - why is there a dark area labelled 'electrode' that also disappears?

Reviewer #2

(Remarks to the Author)

This work describes a self-healable plastic ceramic electrolyte for high-performance and long-life solid-state batteries. The authors employed a dynamically crosslinked polymer with non-covalent -CH₃...CF₃ interaction. The interaction enables self-infiltration of the polymer, resulting in high mechanical stability and ionic conductivity. The materials properties, self-infiltration mechanism, and battery performance were well-characterized. Especially, their direct visualization of polymer and LAMP migration through operando XRF characterization is very impressive. Their dual protection strategy improves the full cell performance despite the facile fabrication. Thus, the reviewer recommends publication of this manuscript in Nature Communications after addressing following issues through a minor revision.

Comments

1. The dynamic interaction that the authors employed is $-\text{CH}_3\cdots\text{CF}_3$ interaction which is not typical hydrogen bonding. The CH_3 group here is only partially polarized through inductive effect of ester oxygen atom. The authors provide ^1H NMR spectra as experimental evidence of hydrogen bonding. However, the peak shift is very small (< 0.2 ppm). In addition, the formation of hydrogen bonding typically makes the proton in the hydrogen bonding donor downfield-shifted, while the proton in EA of the polymer is upfield-shifted. The effect of hydrogen bonding on the chemical shift is concentration-dependent, but the concentrations of the NMR samples are not provided. How strong is this interaction? Please clarify this by providing clearer experimental evidence like FT-IR and concentration-dependent NMR studies, or literature examples that utilize this interaction for dynamic crosslinking (currently only one reference is provided.).
2. In Figure 1a, the schematic representation of the polymer ceramic network seems like having covalent crosslinkers in the polymer backbone (black line). According to the synthetic procedure, however, no crosslinker was used. Please clarify.
3. In Figure 1c, the authors provided the photographic images of bulk PCE. Considering that the $-\text{CH}_3\cdots\text{CF}_3$ interaction is weak, the polymer (SH-SPE) should be very soft. What is the difference in the mechanical properties (modulus and stretchability) of SH-SPE and PCE?
4. There are many previous reports regarding SSLMB composed of PEG-based polymer electrolytes. Can authors provide performance comparison (e.g. durability, capacity retention) with PEG-based polymer electrolytes?
5. Please indicate chemical bonds (e.g. $-\text{CH}_3\cdots\text{CF}_3$) by en dash ($-$). In line 375, ' $-\text{CH}_3$ ' is indicated by en dash, but ' $-\text{CF}_3$ ' is indicated by hyphen. Most of the chemical bonds in the manuscript is indicated by hyphen as well.

Reviewer #3

(Remarks to the Author)

The authors investigated a cold-milled plastic ceramic electrolyte strategy, which embedding a commercial $\text{Li}_{1.5}\text{Al}_{1.5}\text{Ti}_{1.5}(\text{PO}_4)_3$ powder into a self-healing solid polymer electrolyte network. This strategy has simultaneously addressed the conductivity, interface, mechanical, stacking pressure, and fabrication challenges of solid-state LiO -anode batteries and demonstrated long cycle life, high current density, and high area capacity full cells. The authors also used operando XRF, cryo-TEM and ^6Li - ^6Li 2D exchange NMR to explain the principle of self-healing and the way lithium ions are transported through the electrolyte. Here are some comments for this manuscript:

1. The plastic ceramic electrolyte (PCE) by embedding a commercial $\text{Li}_{1.5}\text{Al}_{1.5}\text{Ti}_{1.5}(\text{PO}_4)_3$ (LATP, 70 wt%) powder into a solid polymer electrolyte (SH-SPE, 30 wt%). Why are large quantities of polymers being used, and does the use of large quantities of polymers affect the advantages of solid-state electrolytes such as low-temperature performance, safety, energy density, etc.?
2. As described in Fig. 1b, the NMR shift upfield has been used to explain interactions between monomers. In NMR testing, the mixing of different substances always causes some signal shift. Are these data sufficient to support the authors' theory about dynamic chemical bonding. Are there other characterization data to accompany the evidence.
3. As described in Fig. 1a, the authors attribute the self-healing ability of plastic ceramic electrolyte to the non-covalent interaction between $-\text{CH}_3\cdots\text{CF}_3$. Such interactions include mainly those. If only weak interaction forces, such as van der Waals forces, are sufficient to support their self-healing.
4. As described in Fig. 2a and Fig. 2c, in the operando XRF images, the sulfur element in SPE (highlighted in green) and the phosphorus element in LATP (highlighted in red) are unevenly distributed, does this uneven distribution affect the homogeneity of the use.
5. As described in Fig. 2c and Fig. 2d, the authors find that the self-healing process significantly accelerated with decreased void size. What caused this.
6. As described in Fig. 2c, the two $300\text{-}\mu\text{m}$ -sized voids were completely self-repaired within 20 hours. Are there advantages over other self-healing classes of solid electrolytes. The self-healing process requires in 0.5 C and 20 hours cycling, whether it is feasible in practical applications.
7. As described in Fig. 4, the authors find that both the polymer phase and ceramic phase serve as effective ion conduction pathways. However, due to the difference in ionic conductivity between the polymer phase and the ceramic phase, does it affect the transport of lithium ions, leading to non-uniform deposition and the creation of dendrites.
8. As described in Fig. 6f and 6g, what might be the reason for the lower efficiency of the battery during the first 5 to 10 laps, this phenomenon does not seem to occur in the LFP battery system.
9. As described in Fig. 6g, the H-SSE was paired with a high-Ni, zero-Co, zero-strain cathode in coin cells. The zero-strain test is more meaningful in pouch cells than in coin cells. What happens in pouch cells.

10. In the synthesis of the plastic ceramic electrolyte, a large number of polymer electrolytes are used. What are the advantages of this composite plastic ceramic electrolyte over traditional polymer electrolytes.

Reviewer #4

(Remarks to the Author)

Reviewer #5

(Remarks to the Author)

Version 1:

Reviewer comments:

Reviewer #1

(Remarks to the Author)

Many thanks for the detailed response to reviewers' comments - the paper is now fit for publication from my side

Reviewer #2

(Remarks to the Author)

I thank the authors for carefully addressing my comments, performing additional experiments to address the concerns and revising their manuscript accordingly. I believe the manuscript has been significantly improved and can be accepted for publication.

Reviewer #3

(Remarks to the Author)

My questions have been well responded to. I am happy to recommend the publication of this work.

Reviewer #4

(Remarks to the Author)

Reply to Reviewers' comments

This manuscript has been reviewed by five reviewers. We are grateful that all referees recognize the value and impact of our work. In the meantime, the referees raised several concerns that need to be addressed or clarified. In the past 8 weeks, we have designed new experiments, allocated a lot of resources for these experiments, and obtained new results to support our claims and to fully address the reviewers' concerns. The major revisions are highlighted (in red) in the revised text. A point-by-point response is provided in this letter.

Reviewer #1 (Remarks to the Author):

This is a comprehensive and well presented set of work focused on synthesis of novel self healing polymers for ceramic Li⁺ conducting electrolytes and includes a high level of characterisation and analysis. The cycling and lifetime performance achieved through these material innovations is impressive and the conclusions drawn are largely valid from the data presented. I suspect the work will be impactful in the field if published.

I have returned a word copy of the document including tracked changes for minor corrections suggestions, and comments where I have questions or require more clarity. These comments are also summarised as below:

Reply: Thank you for your valuable suggestions and positive feedback. We also appreciate your careful editing of our manuscript, which greatly improves the quality of our work. We have attached a <Manuscript-with annotations> file in response to your corrections and annotations. Following your suggestions, we have meticulously revised the manuscript, and we believe it now meets the high standards of Nature Communications.

1) Figure 1: It's not clear what the lighter and darker materials are in these photos, and if the 'self healed' has formed from the crack in the top picture, and if so, how. Some labelling and increased info in the caption would be useful here.

Reply: Thank you for your constructive suggestion. The lighter material is the plastic ceramic electrolyte (PCE), while the darker material is the background. The self-healed PCE is formed from the crack in the top picture. In response to your suggestion, we have added labelling to Figure 1c (Figure R1) and provided more information in the figure caption as follows:

Page 4, lines 5 to 7:

“Photographs showing the self-healing ability and flexibility of PCE at room temperature. After cracking, as shown in the top picture, the PCE can self-heal after hand-milling at room temperature for 1 minute, as shown in the middle picture.”

Figure R1 (Figure 1c). Photographs showing the self-healing ability and flexibility of PCE at room temperature. After cracking, as shown in the top picture, the PCE can self-heal after hand-milling at room temperature for 1 minute, as shown in the middle picture.

2) What is the driving force for migration of LATP particles to fill the voids? Why would this happen?

Reply: Thank you for your insightful questions. We propose that the migration of LATP particles within the polymer network occurs via a diffusion process. Initially, the polymer component infiltrates the voids as the first step of the two-step self-healing mechanism, resulting in a significantly lower LATP concentration in the newly repaired area compared to the bulk electrolyte. This concentration gradient further drives the diffusion of LATP ceramics into the voids. Unlike previous composite electrolyte systems that utilize crosslinked or crystallized polymers, our study employs a dynamic polymer with reversible non-covalent bonding. This characteristic further facilitates the diffusion of ceramic particles, thereby enhancing the self-healing process. We appreciate your constructive feedback and have revised the manuscript accordingly to incorporate your valuable suggestions.

Page 5, lines 45 to 47:

“To investigate the effect of cycling current density on the self-healing rate, we provided XRF images illustrating the self-healing of 140- μm -sized void at a cycling current density of 0.05 mA/cm² (Figure S29-31). At this lower current density, the self-healing rate was 23.6 $\mu\text{m}/\text{hour}$ for 142- μm voids and 50.9 $\mu\text{m}/\text{hour}$ for 76- μm voids. For comparison, at 0.2 mA/cm², the self-healing rate was 22.6 $\mu\text{m}/\text{hour}$ for voids sized of 226 μm (Figure 2). **These findings suggest that the self-healing rate remains relatively constant across different cycling conditions, indicating that the process is primarily diffusion-limited. During the first step of self-healing, the polymer component infiltrates the voids, creating a concentration gradient that drives the diffusion of LATP ceramics into the voids to facilitate repair.**”

3) Does the speed of self healing (20 hours, page 4) depend on the cycling or plating/stripping rate? Or is it more like a diffusion limited process?

Reply: We appreciate your professional suggestions. To investigate the impact of cycling current density on the self-healing rate, we examined the self-healing process when cycling the *in-situ* cell at a lower current density of 0.05 mA/cm² (Figure R2-R3), and quantified the self-healing rate by measuring the area of residual voids over time. As shown in Figure R4, at 0.05 mA/cm², the self-healing rate is 23.6 $\mu\text{m}/\text{hour}$ for 142- μm void and 50.9 $\mu\text{m}/\text{hour}$ for 76- μm void. For comparison, at 0.2 mA/cm², the self-healing rate is 22.6 $\mu\text{m}/\text{hour}$ for void size of 226 μm (Figure 2). These results indicate that the cycling condition of the *in-situ* cell does not significantly impact the self-healing rate. This finding aligns with the discussion in our response to Question 2, suggesting that the self-healing process is more characteristic of a diffusion-limited process, where the concentration gradient serves as the primary driving force. Thank you again for your constructive suggestions. We have revised the manuscript on page 5, lines 41-45 to incorporate your valuable inputs.

“...the self-healing rate increases from ~5.6 to ~22.6 $\mu\text{m}/\text{hour}$, which is much faster than the Li⁰-deposition speed at 1 mA/cm² (4.82 $\mu\text{m}/\text{hour}$). **To investigate the effect of cycling current density on the self-healing rate, we provided XRF images illustrating the self-healing of 150- μm -sized void at a**

cycling current density of 0.05 mA/cm^2 (Figure S29-31). At this lower current density, the self-healing rate was $23.6 \text{ }\mu\text{m/hour}$ for $142\text{-}\mu\text{m}$ void and $50.9 \text{ }\mu\text{m/hour}$ for $76\text{-}\mu\text{m}$ void. For comparison, at 0.2 mA/cm^2 , the self-healing rate was $22.6 \text{ }\mu\text{m/hour}$ for void size of $226 \text{ }\mu\text{m}$ (Figure 2). These findings suggest that the self-healing rate remains relatively constant across different cycling conditions, indicating that the process is primarily diffusion-limited. During the first step of self-healing, the polymer component infiltrates the voids, creating a concentration gradient that drives the diffusion of LATP ceramics into the voids to facilitate repair.”

Figure R2 (Figure S29). Overlaid S K-edge and P K-edge XRF mappings of the PCE-based *in-situ* cell when cycling at 0.05 mA/cm^2 for 6 hours.

Figure R3 (Figure S30). Individual P K-edge XRF mappings of the PCE-based *in-situ* cell when cycling at 0.05 mA/cm^2 for 6 hours.

Figure R4 (Figure S31). Size evolution of the 150- μm -sized void when cycling the *in-situ* cell at 0.05 mA/cm².

4) The improved durability (page 11) is attributed to high modulus of ceramic components. But pure OCEs have high modulus and still suffer dendrites and cracking - how is this explained? Perhaps it is not the high modulus of the ceramic component, but the combination of the ceramic with the flexible polymer ‘filler’ in between that does not provide sites for Li plating and therefore dendrites/shorts through the electrolyte.

Reply: Thank you for your professional suggestions. We concur with the reviewer that the synergy between the ceramic and polymer electrolytes contributes to the improved durability. While ceramic electrolytes possess a high modulus theoretically capable of blocking dendrite penetration, the voids and grain boundaries present in ceramics can act as sites for Li⁰ plating, leading to dendrite formation. Furthermore, the non-intimate interfacial contact between the Li⁰ anode and ceramics can result in high local current densities, which further promote dendrite growth [Nature Materials, 2021, 20(4): 503-510]. In this study, the polymer component infiltrates these voids and grain boundaries, effectively eliminating “hot-spots” for dendrite formation. Additionally, the polymer component provides more intimate interfacial contact, reducing local current concentration and enabling stable Li⁰ deposition [Nature Materials, 2022, 21(9): 1050-1056]. In response to your suggestion, we have revised the manuscript accordingly on page 12, lines 7 to 11 as follows:

“This dramatically improved durability of PCE could be attributed to the combination of the ceramic with the flexible polymer filler. The polymer component can infiltrate voids and grain boundaries, eliminating 'hot-spots' for dendrite formation. Additionally, the polymer component provides more intimate interfacial contact, minimizing local current concentration and enabling stable Li⁰ deposition.”

5) Figure 5 a - How many cycles does the time correspond to? Can this be added to the figure as a label or in the caption?

Reply: We appreciate your constructive suggestions. The cycling life of the Li⁰/LATP/Li⁰, Li⁰/SH-SPE/Li⁰, and Li⁰/PCE/Li⁰ cells are 90, 200, and 860 cycles, respectively. We have added these cycling numbers to Figure R5 (Figure 5a), as shown below. Thanks for your valuable input.

Figure R5 (Figure 5a). Durability comparison of LAMP, SH-SPE, and PCE under $\text{Li}^0\text{-Li}^0$ cell configuration.

6) Page 13 some ‘cycling’ mentions only an accumulated capacity and some mentions a cycle capacity, which leads me to think that some of this cycling is ‘charge discharge’ to some set capacity limit, and some is ‘one way’ to a total plating capacity. But it’s not clear why, if so, there are different protocols. If so, clarify if this is different to the 1 mA/cm^2 and 2900 mAh/cm^2 AAC in the previous experiment

Reply: Thank you for your professional questions. In Figure 5a and Figures S23-S27, a single testing protocol is utilized, which involves cycling rather than unidirectional plating. The cell is charged at a certain current density (e.g., 0.2 mA/cm^2 in Figure 5a) to a set capacity limit (e.g., 0.5 mAh/cm^2 in Figure 5a), and then discharged at the same current density to the same capacity limit. This charge-discharge cycle is then repeated, with cycling life in hours shown on the x-axis and cell voltage on the y-axis. This testing protocol is generally adopted in the field of solid-state batteries [Nature Nanotechnology, 2023, 18(6): 602-610; Nature nanotechnology, 2022, 17(9): 959-967], allowing us to evaluate the durability of our electrolytes against Li^0 anode.

In addition, the “accumulated areal capacity (AAC)” refers to the total amount of plated Li^0 during cycling, calculated as $\text{AAC} = \text{Current density (mA/cm}^2) \times \text{cycling time (hours)}$ [Advanced Materials 31.3 (2019): 1804815]. For example, in Figure S23A, the cell was cycled at 1 mA/cm^2 for 2900 hours, resulting in an AAC of 2900 mAh/cm^2 . To avoid misunderstanding, we have carefully defined the AAC in the revised manuscript (please refer to the response to question 6B).

Additionally, we have addressed your three annotations in the word copies regarding similar concerns in this response letter (Please see questions 6B, 6C, and 6D). Once again, thank you for your careful review and constructive suggestions, which greatly enhance the quality of our work.

6B) Page 12, This isn’t cycling though, is it? It’s Li plating in one direction. Change to e.g. ‘stable Li plating’

Reply: Thank you for your insightful question, and we apologize for the misunderstanding. Figures S23-S27 illustrate the cycling of $\text{Li}^0\text{-Li}^0$ cells, not Li^0 plating in one direction. The detailed testing protocol is described above. To prevent any confusion, we have revised the text as follows:

Page 14, lines 3 to 7:

“The $\text{Li}^0/\text{PCE}/\text{Li}^0$ cell was then cycled at current density of 1 mA/cm^2 , demonstrating stable operation for 2900 hours (Figure S23). The total amount of plated Li^0 during cycling can be quantified by the accumulated areal capacity (AAC), calculated as $\text{AAC} = \text{Current density (mA/cm}^2) \times \text{Cycling time}$

(hours). Consequently, when cycling at 1 mA/cm², the AAC of the Li⁰/PCE/Li⁰ cell reached 2900 mAh/cm².”

6C) What is this referring to? Areal loading, in a symmetric cell, with unidirectional plating? Or is this now actually cycling, to that capacity before the charging/discharging direction is changed? If so, clarify if this is different to the 1 mA/cm² and 2900 mAh/cm² AAC in the previous experiment

Reply: Thank you for your constructive suggestion. The “2 mAh/cm²” refers to the cut-off capacity before the charging/discharging direction is changed. This cycling protocol is the same as the previous experiment, which cycles the Li⁰-Li⁰ symmetric cell at 1 mA/cm². To avoid misunderstanding, we have revised the manuscript as follows:

Page 14, lines 7 to 9:

“When cycling the cell at a higher current density of 2 mA/cm² and a large cut-off areal capacity of 2 mAh/cm², the cell also delivered an AAC of 2000 mAh/cm² (Figure S24).”

6D) Again, it’s not clear if this is unidirectional plating without cycling, or if it is cycling to some cut off capacity. Only the second of the 4 current densities mentioned here has a specified areal capacity (not AAC) - is this the only experiment that used cycling, and the others were continuous plating/stripping (unidirectional)? If so, what is the rationale for this?

Reply: Thank you for your constructive suggestion. Figures S25-S27 also showed the cycling of Li⁰/PCE/Li⁰ cells (not unidirectional plating). At current densities of 5 mA/cm², 10 mA/cm², and 20 mA/cm², the cut-off areal capacities are 2 mAh/cm², 1 mAh/cm², and 0.5 mAh/cm², respectively. In response to your suggestions, we have provided the cut-off areal capacity during cycling in the revised manuscript. Thank you again for your insightful input.

Page 14, lines 9-13:

“In addition, we further cycled the Li⁰/PCE/Li⁰ cells at higher current densities of 5 mA/cm², 10 mA/cm², and 20 mA/cm², with cut-off areal capacities of 2 mAh/cm², 1 mAh/cm², and 0.5 mAh/cm², respectively. As shown in Figures S25-S27, the AAC reached 1400 mAh/cm² and 1500 mAh/cm² when the cells were cycled at 20 mA/cm² and 10 mA/cm², respectively.”

7) Page 13 - Why is the rate used for NMC811 so much lower than for LFP? The initial capacity is also lower than expected for 811. Could the authors note and explain this in the text?

Reply: We appreciate your insightful suggestions. The Li⁰-NMC811 cell was cycled at a lower C-rate due to the high loading of the NMC811 cathode and the operation of the cell at room temperature. Specifically, the areal loading of the LFP cathode is ~0.4 mAh/cm², with C/2 corresponding to a current density of 0.2 mA/cm². In comparison, the areal loading of the NMC811 cell is 1.6 mAh/cm², with C/6 corresponding to a current density of 0.27 mA/cm². This current density is comparable to that of the Li⁰-LFP cell when operating at room temperature. We also acknowledge your observation regarding the lower-than-expected initial capacity of the Li⁰-NMC811 cell. This phenomenon could be attributed to possible side reactions between PCE and NMC811 at high voltages. Additionally, the higher bulk resistance and charge transfer resistance at room temperature may increase the overpotential, resulting in voltage hysteresis that limits the discharge capacity of the NMC811 cathode [Advanced Materials,

2017, 29(22): 1606042]. In future studies, the discharge capacity could be enhanced by developing a well-optimized CEI chemistry to improve electrochemical stability when using high-voltage cathodes [Nano Energy, 72 (2020), 104655]. Moreover, fabricating composite cathodes that contain polymer or inorganic electrolytes could improve ion conduction within the electrode [Energy Environ. Sci., 2020, 13, 908-916], potentially leading to improved capacity. In response to your suggestion, we have incorporated and explained this in the revised manuscript as follows:

Page 14, lines 25 to 26:

“Due to its high loading (1.6 mAh/cm^2), the r.t. performance of the Li^0 -NMC811 cell was evaluated at a lower C-rate of C/6.”

Page 14, lines 31 to 37:

“Despite demonstrating good cycling durability, the initial low coulombic efficiency and discharge capacity need further improvement. These issues may stem from potential side reactions at the cathode-electrolyte interface and high resistance at room temperature, which contributes to voltage hysteresis⁴⁰. Future studies will focus on enhancing the initial coulombic efficiency by developing a stable cathode-electrolyte interface (CEI)⁴¹ and addressing the low discharge capacity through the design of composite cathodes to improve ion conduction within the electrode.”

8) Figure 6g - Is the efficiency in this figure referring to the H-SSE or PA-SPE? Specify, and add in the other if missing. Change ‘Efficiency’ to Coulombic Efficiency’ on axis labels for these graphs

Reply: We appreciate your constructive suggestions. The efficiency in Figure 6g refers to H-SSE, and we have revised the figure legend to specify this. Additionally, we have provided the Coulombic efficiency of PA-SPE in Figure R6. Furthermore, we have changed all “Efficiency” to “Coulombic Efficiency” as the y-axis labels for graphs showing the full cell performances. Thank you again for your valuable input!

Page 16, lines 3 to 6, caption for Figure 6g:

“g, Performances of H-SSE-based and PA-SPE-based full cells when employing a high-Ni, zero-Co, zero-strain cathode. The right y-axis displays the Coulombic efficiency of the H-SSE-based full cell, while the Coulombic efficiency of the PA-SPE-based full cell is shown in Figure S32.”

Figure R6 (Figure S32). Performance of PA-SPE-based full cells when employing the high-Ni, zero-Co, zero-strain cathode. The left y-axis displays the discharge capacity, while the right y-axis represents the Coulombic efficiency of the cell.

9) The high loading NMC cathode - Was this provided as pre-cast electrodes, and if so, how did you ensure electrolyte infiltration/mixing with the cathode? Was a composite cathode/SSE made?

Reply: Thank you for your professional suggestion. The high-loading NMC811 cathode was provided as pre-cast electrode sheets, comprising 90 wt% active material, 5 wt% conductive carbon, and 5 wt% PVDF binder. In this study, we did not fabricate a composite cathode containing SSE. To verify electrolyte infiltration and mixing with the cathode, we have included SEM-EDS mapping of the cross-section of an NMC811 cathode after cycling with H-SSE. As shown in Figure R7, S and N elements from the polymer component are distributed homogeneously within the cathode. This suggests that lithium salt (LiTFSI) and succinonitrile solid crystal plasticizer infiltrate the cathode layer's pores, creating an ion conduction pathway. Consequently, H-SSE can cycle with commercial pre-cast cathodes without the need for additional ceramic or polymeric ionic conductors, thereby enhancing the compatibility of our electrolyte with conventional battery fabrication techniques. In response to your suggestion, we have revised the manuscript on page 18, lines 25 to 32 to incorporate your valuable input:

“To confirm electrolyte infiltration and mixing with the cathode, SEM-EDS mapping of the cross-section of an NMC811 cathode after cycling with H-SSE is provided in Figure S33. N and S elements from the polymer component were found to be homogeneously distributed within the cathode. This suggests that lithium salt (LiTFSI) and succinonitrile solid crystal plasticizer could infiltrate the pores of the cathode layer, creating an ion conduction pathway. As a result, H-SSE can cycle with commercial pre-cast cathodes without the need for additional ceramic or polymeric ionic conductors, enhancing the compatibility of our electrolyte with conventional battery fabrication techniques.”

Figure R7 (Figure S33). SEM-EDS mapping of the cross-section of the NMC811 cathode after cycling with H-SSE. The homogeneous distribution of N and S elements within the cathode indicates successful infiltration of lithium salt (LiTFSI) and succinonitrile solid crystal plasticizer into the cathode pores, thereby creating an ion conduction pathway.

10) Figure S4 – why is there a dark area labelled ‘electrode’ that also disappears?

Reply: Thank you for your valuable question. Figure S4 presents XRF images of the interface between the PCE and the stainless steel (SS) working electrode. As depicted in the tube geometry of the *in-situ* cell (Figure R8), the diameter of the SS electrode is slightly smaller than the inner diameter of the

Kapton® tube to facilitate the insertion of the electrode into the tube. This creates a small gap between the SS electrode and the Kapton® tube. Over cycling time, the PCE infiltrates this gap, similar to how it infiltrates voids during the self-healing process. Consequently, the electrode becomes obscured by the infiltrated PCE. In response to your suggestion, we have revised the caption of Figure S4 to clarify this explanation. Thank you again for your constructive input.

Page 4, lines 6-12 on supporting information:

“**Figure S4.** High-resolution XRF images of the interface between the PCE and the stainless steel (SS) working electrode. As illustrated by the tube geometry of the *in-situ* cell (Figure 2b), the SS electrode's diameter is slightly smaller than the inner diameter of the Kapton® tube, allowing for its insertion into the tube. This creates a small gap between the SS electrode and the Kapton® tube. Over cycling time, PCE infiltrates this gap in a manner similar to its infiltration into voids during the self-healing process. Consequently, the electrode becomes obscured by the infiltrated PCE. This observation suggests good wettability and interface compatibility of the PCE with the electrode.”

Figure R8 (Figure 2b). A schematic illustration of the tube battery and experimental set-up for XRF and XAS characterization.

Reviewer #2 (Remarks to the Author):

This work describes a self-healable plastic ceramic electrolyte for high-performance and long-life solid-state batteries. The authors employed a dynamically crosslinked polymer with non-covalent $-\text{CH}_3\cdots\text{CF}_3$ interaction. The interaction enables self-infiltration of the polymer, resulting in high mechanical stability and ionic conductivity. The materials properties, self-infiltration mechanism, and battery performance were well-characterized. Especially, their direct visualization of polymer and LATP migration through operando XRF characterization is very impressive. Their dual protection strategy improves the full cell performance despite the facile fabrication. Thus, the reviewer recommends publication of this manuscript in *Nature Communications* after addressing following issues through a minor revision.

Reply: Thank you for your thorough review and positive feedback on our manuscript. We appreciate your constructive suggestions, which have greatly contributed to the improvement of our work. We have revised the manuscript in accordance with your comments, as detailed below:

Comments

1. The dynamic interaction that the authors employed is $-\text{CH}_3\cdots\text{CF}_3$ interaction which is not typical hydrogen bonding. The CH_3 group here is only partially polarized through inductive effect of ester oxygen atom. The authors provide ^1H NMR spectra as experimental evidence of hydrogen bonding. However, the peak shift is very small (< 0.2 ppm). In addition, the formation of hydrogen bonding typically makes the proton in the hydrogen bonding donor downfield-shifted, while the proton in EA of the polymer is upfield-shifted. The effect of hydrogen bonding on the chemical shift is concentration-dependent, but the concentrations of the NMR samples are not provided. How strong is this interaction? Please clarify this by providing clearer experimental evidence like FT-IR and concentration-dependent NMR studies, or literature examples that utilize this interaction for dynamic crosslinking (currently only one reference is provided.).

Reply: We appreciate your professional comments and suggestions. We agree that hydrogen bonding typically refers to the dipole-dipole attractions between a proton acceptor and a proton donor that contains N-H , O-H , or F-H bonds. In the revised manuscript, we have reclassified the $-\text{CH}_3\cdots\text{CF}_3$ interaction as a non-covalent interaction. We also acknowledge that the formation of such non-covalent interaction typically causes downfield shifting of proton signals from the proton donor. In the previous version of the manuscript, we used liquid-state NMR to investigate the non-covalent interactions between ethyl acrylate (EA) and MTFSI monomers. The S=O polar group in the deuterium solvent (DMSO-d_6) might also interacts with the EA monomers, potentially interfering with the NMR results. In this revised manuscript, we have employed Magic Angle Spinning Solid-State NMR (MAS-ssNMR) to reveal the $-\text{CH}_2-\text{CH}_3\cdots\text{CF}_3$ non-covalent bonding. No deuterium solvent that could interfere with the NMR results was used, and all spectra were calibrated against trimethylsilane (TMS, 0 ppm). Additionally, in response to your suggestion, we have conducted concentration-dependent NMR studies to provide clearer experimental evidence. As shown in Figure R9, the pristine EA (i.e. EA:MTFSI= 4:0, w/w) shows ^1H signals at 0.93, 3.84, and 5.79 ppm, corresponding to the $-\text{CH}_3$, $-\text{CH}_2-$, and $\text{CH}_2=\text{CH}$ groups, respectively. With increasing MTFSI content, all signals of EA were found to gradually shift downfield (Figure R10), which could be due to the electron-withdrawing effect of F or O atoms in MTFSI, causing a decrease in the electron cloud density of EA molecules. Consistent

with these observations, as the EA content in MTFSI increased, the signals of MTFSI gradually shifted upfield (Figure R11), confirming the existence of non-covalent interactions between EA and MTFSI.

Regarding the intensity of the above non-covalent interactions, Jung and co-workers employed DFT calculations and reported that the binding energy between TFSI⁻ and the -CH₂- group in a polyester backbone is -0.457 eV [Adv. Mater. 2018, 30, 1706851]. Similarly, Wang et al. used molecular dynamics simulations and reported that the binding energy between the FSI⁻ anion and the -CH₂- group in the PEO backbone is -14.6 kcal/mol (-0.62 eV) [Nature Materials, 21(9), 1057-1065], which aligns with Jung's findings. For comparison, the strength of water-water hydrogen bonding has been reported to be -0.25 eV [Physical Review B, 2006, 74(24): 245409], which is lower than the binding energy of the aforementioned non-covalent interactions. Therefore, this polymer-anion interaction is widely applied for dynamic crosslinking to prepare functional polymers. For example, Cao and co-workers applied this non-covalent interaction between the poly(ethyl acrylate) backbone and TFSI⁻ anion to prepare tough ionogels [ACS Appl. Polym. Mater. 2020, 2, 2359-2365]. Liu's group employed this non-covalent interaction between TFSI⁻ and the -CH₂- group in the polymer backbone to synthesize robust and stretchable elastomers [Materials Horizons, 2020, 7(3): 912-918].

Once again, we greatly appreciate your constructive suggestions, which have deepened our understanding of polymer-anion non-covalent interactions. We have carefully revised our manuscript accordingly on Page 2, lines 23-33, as follows:

“As evidence, the magic angle spinning solid-state NMR (MAS-ssNMR) spectra presented in Figure 1b and Figure S1 demonstrate that as the (trifluoromethane) sulfonimide lithium methacrylate (MTFSI) content increases, the signals of ethylene acrylate (EA) shift gradually downfield. This shift can be attributed to the electron-withdrawing effects of the F/O atoms in MTFSI. Conversely, with increasing EA content, the signals of MTFSI shift upfield (Figure S1c), which further supports the presence of non-covalent interactions between EA and MTFSI. Previous theoretical predictions also show a high binding energy of 0.4-0.5 eV for this -CH₃...CF₃ interaction^{20,21}, which surpasses the strength of water-water hydrogen bonding (0.25 eV)²². Consequently, this non-covalent interaction is extensively utilized in developing functional polymers, including mechanically robust ionogels²³ and stretchable elastomers²⁴. In this work,...

Figure R9 (Figure S1a). Full range Magic Angle Spinning Solid-State NMR (MAS-ssNMR) spectra of EA/MTFSI mixture at different mass ratio. The spectra were referenced against trimethylsilane (TMS, 0 ppm). The spinning rate of sample is 2000 Hz.

Figure R10 (Figure 1b and Figure S1b). Enlarged MAS-ssNMR spectra showing the downfield shifting of EA signals with increasing MTFTSI content.

Figure R11 (Figure S1c). Enlarged MAS-ssNMR spectra showing the upfield shifting of MTFSI signals with increasing EA content.

2. In Figure 1a, the schematic representation of the polymer ceramic network seems like having covalent crosslinkers in the polymer backbone (black line). According to the synthetic procedure, however, no crosslinker was used. Please clarify.

Reply: Thank you for bringing this to our attention. We did not employ any covalent crosslinker when synthesizing the polymer backbone. In response to your suggestion, we have revised Figure 1a (Figure R12) and removed the black line indicating covalent crosslinking. Corresponding revisions have been made on page 16, lines 40-41 of the manuscript:

“Note that no covalent crosslinker was employed to ensure the efficient infiltration of polymer into the voids/cracks.”

Figure R12 (Figure 1a). Schematic illustration showing the design concept of PCE through embedding the LATP powder into a dynamic polymer network.

3. In Figure 1c, the authors provided the photographic images of bulk PCE. Considering that the $-\text{CH}_3 \cdots \text{CF}_3$ interaction is weak, the polymer (SH-SPE) should be very soft. What is the difference in the mechanical properties (modulus and stretchability) of SH-SPE and PCE?

Reply: Thank you for your constructive suggestions. In this revised manuscript, we provided the modulus of SH-SPE and PCE measured by a rheometer. Figure R13 shows the viscoelastic properties

of SH-SPE, illustrating the change in storage modulus (G') and loss modulus (G'') with increasing strain rate. The SH-SPE exhibits a G' range from 10^4 to 10^5 Pa. Across the changing strain rates, G' consistently exceeds G'' , confirming the solid nature of SH-SPE [Nature Energy, 2019, 4(5): 365-373]. Upon introducing 70 wt% LATP ceramic, G' dramatically increased to 10^6 - 10^7 Pa, which could be due to the particle-particle friction enhancing the mechanical rigidity of the PCE [Progress in Polymer Science 75 (2017) 48–72; Reports on Progress in Physics 77.4 (2014): 046602]. Figure R14 depicts the stretchability of SH-SPE and PCE. SH-SPE can be stretched to over 405% of its original length without breaking. In comparison, PCE exhibits reduced stretchability due to the ceramic particles but can still stretch to 220% of its original length without breaking. This good flexibility of PCE, distinct from conventional ceramic electrolytes, is beneficial for avoiding mechanical fractures during battery fabrication and operation. Once again, we appreciate your valuable suggestions and have revised the manuscript on page 17, lines 22 to 30 to incorporate your thoughtful inputs.

“The modulus of SH-SPE and PCE was measured using a rheometer. Figure S34 shows the viscoelasticity of SH-SPE, indicating the change in storage modulus (G') and loss modulus (G'') with increasing strain rate. SH-SPE exhibits a G' range from 10^4 to 10^5 Pa. As the strain rate changes, G' consistently exceeds G'' , confirming the solid nature of SH-SPE⁴⁵. With the introduction of 70 wt% LATP ceramic, G' dramatically increases to 10^6 - 10^7 Pa, which could be due to particle-particle friction enhancing the mechanical rigidity of PCE^{46,47}. Figure S35 also illustrates the stretchability of SH-SPE and PCE, both of which can be stretched to over 220% of their original length without breaking. This suggests good flexibility, which is ideal for avoiding mechanical fractures during battery fabrication and operation.”

Experimental section, page 19, lines 9-10:

“The viscoelastic properties of SH-SPE and PCE were evaluated using a TA DHR-2 rheometer in oscillation mode with a parallel plate configuration.”

Figure R13 (Figure S34). Viscoelasticity of SH-SPE (a) and PCE (b) showing the storage modulus (G') and loss modulus (G'') with increasing strain rate.

Figure R14 (Figure S35). Photographs showing the stretchability of SH-SPE and PCE.

4. There are many previous reports regarding SSLMB composed of PEG-based polymer electrolytes. Can authors provide performance comparison (e.g. durability, capacity retention) with PEG-based polymer electrolytes?

Reply: Thank you for your constructive suggestions. We have summarized the performance of previously reported PEG-based polymer electrolytes in Table R1. In $\text{Li}^0\text{-Li}^0$ symmetric cells, most documented solid polymer electrolytes (SPEs) were cycled at low current densities, typically ranging from 0.1 to 1 mA/cm^2 , to mitigate dendrite formation. These cells generally exhibit a cycling life of 100 to 1000 hours, resulting in a low accumulated areal capacity (AAC) of 50 to 500 mAh/cm^2 . In contrast, our study demonstrates stable cycling of $\text{Li}^0\text{-Li}^0$ cells at higher current densities, ranging from 1 to 20 mA/cm^2 , with AACs of 2900, 1950, 1950, 1500, and 1400 mAh/cm^2 for current densities of 1, 2, 5, 10, and 20 mA/cm^2 , respectively.

For full cells with low mass loading cathodes (2-4 mg/cm^2), the cycling life in these studies were generally below 1000 cycles for low-voltage LFP cathodes and below 300 cycles for high voltage NMC811/ LiCoO_2 cathodes. In comparison, the PCE developed in this study enables extended durability of 4000 cycles with an LFP cathode (88% retention) and 2900 cycles (70% retention) with an 4.2 V zero-strain cathode. For full cells with high mass loading cathodes, Previous PEG-based SPEs generally demonstrated cycling durability of fewer than 100 cycles. Our work, however, shows cycling stability of 640 cycles with a 1.6 mAh/cm^2 NMC811 cathode (80% retention).

Based on the results presented in Table R1, our design shows clear improvements in Li^0 anode compatibility, durability, and area capacity. We believe these results provide substantial evidence of the effectiveness of our self-healing plastic ceramic electrolyte in addressing Li^0 morphological instabilities. Thank you again for your valuable inputs. We have revised the manuscript on page 15, lines 3 to 7 as follows:

“In Table S1-S3, we further compared battery performance with previously reported PEO-based polymer electrolytes. The PCE exhibits notable improvements in Li^0 anode compatibility, durability, and areal capacity, thereby demonstrating the effectiveness of our design in achieving durable and high-performance solid-state batteries.”

Table R1 (Table S1). Comparison of the $\text{Li}^0\text{-Li}^0$ symmetric cell durability and full cell performance with previous Li^0 -anode batteries on PEO-based solid polymer electrolytes.

a) Critical current density; b) 3V-class cathode: LiFePO_4 ; 4V-class cathode: NMC, LiCoO_2 , etc.

References	CCD ^{a)} (mA/cm^2)	$\text{Li}^0\text{-Li}^0$ cell Current/life/temp.	Current× cycle life (mAh/cm^2)	Cathode ^{b)}	Current Density	capacity/cycle life/retention	Loading (mg/cm^2)	Loading (mAh/cm^2) ×cycle number
This work	30	1/2900hr/r.t.	2900	3 V	C/2	127/2400/84%	4.5	1836
		2/1000hr/50°C	2000	3 V	2C	123/4000/88%	2.3	1564
		5/300hr/50°C	1500	4 V	1C	144/2860/80%	3	1844
		10/150hr/50°C	1500	4V	C/6	130/1000/71%	7.4	1600
2024 Nat. Energy ¹	3.7	0.5/1000hr/	500	4V	C/2	210/400/80%	5.3	460
2024 Nat. Nanotechnol. ²	NA	0.1/1000hr/60°C	100	3 V	0.5C	138/200/90%	1.5	51

				3 V	0.07C	130/35/95%	6.8	41
2024 Nat. Commun. ³	0.5	0.1/2250hr/90°C	225	3 V	0.5C	160/1000/78%	1.6	270
2024 Nat. Mater. ⁴	NA	0.05/1500hr/30°C	75	3 V	0.05C	147/40/92%	8.8	53
2024 Angew. Chem. ⁵	1.1	0.4/1500hr/60°C	600	3V	2C	123/1500/80%	2	510
2024 Energy Environ. Sci. ⁶	NA	0.1/100hr/25°C	10	3 V	0.1C	147/100/88%	1.2	21
				4 V	0.1C	152/100/66%	1.2	26
2024 Nano Energy ⁷	0.5	0.1/1600hr/NA	160	4 V	0.1C	166/250/92%	3	161
		0.2/400hr/NA	80	4 V	NA	178/100/82%	10	215
2024 ACS Energy Lett. ⁸	NA	0.4/500hr/70°C	200	3 V	2C	130/1000/76%	2.5	425
2023 J. Am. Chem. Soc. ⁹	NA	0.1/400hr/25°C	40	3 V	0.5C	130/250/96%	2.5	106
				4 V	0.1C	134/100/84%	2.5	42.5
2023 Angew. Chem. ¹⁰	0.3	0.1/1000hr/60°C	100	3 V	0.2C	149/600/75%	1.2	122
2023 PNAS ¹¹	2	0.5/700hr/25°C	350	3 V	0.5C	135/300/92%	2.5	117
				3 V	0.2C	160/100/80%	8	136
2023 Nano Energy ¹²	NA	0.1/2000hr/60°C	200	3 V	1C	127/300/99%	1	51
2023 Nano Energy ¹³	NA	0.5/600hr/NA	300	4 V	0.5C	145/100/92%	2.2	28
2023 Nano Energy ¹⁴	0.7	0.1/900hr/25°C	90	3 V	0.2C	155/330/95%	1.7	95
				4 V	0.1C	152/100/83%	1.1	24
2023 Nano Energy ¹⁵	0.2	0.1/320hr/60°C	32	3 V	0.1C	148/100/93%	NA	NA
2023 Adv. Energy Mater. ¹⁶	NA	0.3/500hr/60°C	150	3 V	1C	147/300/99.7%	2	102
2022 Joule ¹⁷	NA	0.75/1300hr/r.t.	974	3 V	0.25C	160/100/97%	4	68
2022 ACS Energy Lett. ¹⁸	NA	0.1/445hr/70°C	45	4 V	C/20	160/80/80%	4.7	80
2022 Nano Energy ¹⁹	NA	0.2/2400hr/60°C	480	3 V	0.5C	133/1350/83%	3.9	895
2022 Adv. Energy Mater. ²⁰	0.9	0.5/300hr/60°C	150	3 V	0.5C	160/500/70%	3	255
2022 Energy Environ. Sci. ²¹	NA	0.2/2200hr/NA	440	4 V	0.5C	188/1000/80%	2.5	537
2022 Nat. Commun. ²²	NA	0.1/3800hr/90°C	380	3 V	0.5C	158/400/98%	1.5	102
				4 V	0.5C	145/100/87%	1.5	25.2
2022 Nat. Commun. ²³	1.2	0.1/1200hr/NA	120	3 V	0.25C	167/1200/87%	2	480
		1.2/120hr/NA	144					
2022 J. Am. Chem. Soc. ²⁴	NA	0.1/500hr/70°C	50	3 V	C/3	151/200/86%	4	136
2021 J. Am. Chem. Soc. ²⁵	2	0.4/400hr/NA	160	3 V	0.15C	160/120/78%	5	102
		1.2/100hr/NA	120	4 V	0.24C	140/120/86%	3	77.4
2021 Adv. Mater. ²⁶	NA	0.1/1200hr/NA	120	4 V	0.1C	177/100/78%	21	450
		1/1100hr/NA	1100	4 V	0.1C	163/100/89%	23	450
2021 Nano Lett. ²⁷	NA	0.1/300hr/60°C	30	3 V	1C	146/1000/84%	1	170
2020 Adv. Mater. ²⁸	0.5	0.1/1300hr/NA	130	3 V	0.5C	144/500/86%	2.4	204
2020 Energy Environ. Sci. ²⁹	NA	0.2/2500hr/NA	500	3 V	C/3	160/210/97%	3.3	118
				4 V	0.1C	135/110/90%	3.5	82
2020 Nano Lett. ³⁰	NA	0.1/300hr/60°C	30	3 V	0.2C	140/300/95%	1.5	76.5

2020 Nano Lett. ³¹	NA	0.05/600hr/25°C	30	3 V	0.1C	166/250/77%	5	212
2020 Nano Lett. ³²	NA	1/560hr/90°C	560	3 V	1C	140/200/86%	2.5	85
		2/260hr/90°C	520	4 V	C/20	160/50/80%	2.5	27

5. Please indicate chemical bonds (e.g. –CH₃–CF₃) by en dash (–). In line 375, ‘–CH₃’ is indicated by en dash, but ‘–CF₃’ is indicated by hyphen. Most of the chemical bonds in the manuscript is indicated by hyphen as well.

Reply: Thank you for your professional suggestions. We have carefully reviewed the manuscript and standardized the use of en dashes to indicate chemical bonds. The changes have been highlighted in red in the revised manuscript for your reference.

Reviewer #3 (Remarks to the Author):

The authors investigated a cold-milled plastic ceramic electrolyte strategy, which embedding a commercial $\text{Li}_{1.5}\text{Al}_{0.5}\text{Ti}_{1.5}(\text{PO}_4)_3$ powder into a self-healing solid polymer electrolyte network. This strategy has simultaneously addressed the conductivity, interface, mechanical, stacking pressure, and fabrication challenges of solid-state Li^0 -anode batteries and demonstrated long cycle life, high current density, and high area capacity full cells. The authors also used operando XRF, cryo-TEM and ^6Li - ^6Li 2D exchange NMR to explain the principle of self-healing and the way lithium ions are transported through the electrolyte. Here are some comments for this manuscript:

Reply: We sincerely appreciate the meticulous review of our manuscript and the insightful comments provided to enhance its quality. In response, we have extensively revised the manuscript, incorporating additional data to address the raised questions. We believe these constructive additions have significantly strengthened the manuscript. Thank you for your valuable feedback.

1. The plastic ceramic electrolyte (PCE) by embedding a commercial $\text{Li}_{1.5}\text{Al}_{0.5}\text{Ti}_{1.5}(\text{PO}_4)_3$ (LATP, 70 wt%) powder into a solid polymer electrolyte (SH-SPE, 30 wt%). Why are large quantities of polymers being used, and does the use of large quantities of polymers affect the advantages of solid-state electrolytes such as low-temperature performance, safety, energy density, etc.?

Reply: Thank you for your insightful comments. The incorporation of 30 wt% self-healing polymer electrolyte (SH-SPE) is designed to address several key challenges. Firstly, this addition mitigates the high grain boundary resistance of LATP. Specifically, the grain boundary resistance decreases from 5000 ohms to 55 ohms following polymer infiltration (Figure 4f), resulting in a significant enhancement in room temperature ionic conductivity to 0.75 mS/cm (Figure 4g). Secondly, the SH-SPE imparts self-healing properties to the PCE. Operando XRF results demonstrate a dual-phase self-healing process at a rate of 22.6 $\mu\text{m}/\text{hour}$, effectively repairing voids and cracks that could promote dendrite formation (Figure 2). Thirdly, SH-SPE facilitates the formation of a stable solid electrolyte interface (SEI), as supported by *in-situ* XAS and *ex-situ* XPS analyses (Figure 2e-2f and Figure S5-S8). This SEI prevents degradation of LATP and enhances interfacial stability with the Li^0 anode. Lastly, incorporating SH-SPE alleviates the need for high-temperature and high-pressure fabrication processes typical of conventional ceramic electrolytes. Instead, PCE can be prepared via a simple cold-milling process at room temperature, aligning better with current lithium-ion battery manufacturing practices.

We believe that the benefits conferred by the SH-SPE further enhance the advantages of solid-state batteries. The reduction in grain boundary resistance and increased ionic conductivity allow PCE-based Li^0 -anode batteries to operate at higher current density and exhibit lower overpotential compared to pristine LATP electrolyte (Figure 5a), resulting in improved room temperature battery performance. The flexibility and self-healing functionality of PCE reduce mechanical failures and dendrite penetration, thereby decreasing safety hazards associated with short-circuiting. Moreover, the cold-milling process and favorable processability enable the fabrication of thinner PCE layers compared to conventional ceramic electrolytes [e.g., 1000 μm in Nature, 2023, 616(7955): 77-83; 700 μm in Science, 2021, 373: 1494-1499; 780 μm in Nature, 2021, 593: 218-222], which leads to improved

energy density. We have revised the manuscript on Page 17, lines 14 to 21 to incorporate your valuable inputs:

“The incorporation of 30 wt% SH-SPE aims to reduce grain boundary resistance, impart self-healing capabilities, form a stable SEI, and simplify the electrolyte fabrication process. By decreasing grain boundary resistance and enhancing ionic conductivity, the PCE demonstrates improved battery performance at room temperature. The flexibility and self-healing properties of the PCE reduce mechanical failure and dendrite penetration during battery operation, thereby mitigating safety hazards related to short-circuiting. Additionally, the cold-milling process and improved processability enable the fabrication of thinner PCE layers compared to conventional ceramic electrolytes, leading to an increased energy density.”

2. As described in Fig. 1b, the NMR shift upfield has been used to explain interactions between monomers. In NMR testing, the mixing of different substances always causes some signal shift. Are these data sufficient to support the authors' theory about dynamic chemical bonding. Are there other characterization data to accompany the evidence.

Reply: Thank you for your insightful comments. We agree that in liquid-state NMR, the introduction of deuterium solvents (e.g., DMSO-d₆) or variations in sample concentration can lead to signal shifts. To address this concern, we employed Magic Angle Spinning Solid-State NMR (MAS-ssNMR) to investigate the –CH₂–CH₃⋯CF₃ non-covalent bonding. No deuterium solvents that could interfere the NMR result were used, and all spectra were calibrated against trimethylsilane (TMS, 0 ppm). Additionally, to provide more robust experimental evidence for the dynamic bonding between ethyl acrylate (EA) and MTFSI monomers, we conducted concentration-dependent NMR studies. As illustrated in Figure R15, pristine EA (i.e. EA:MTFSI=4:0, w/w) shows ¹H signals at 0.93, 3.84, and 5.79 ppm, corresponding to the –CH₃, –CH₂–, and CH₂=CH groups, respectively. With increasing MTFSI content, the signals of EA shift gradually downfield (Figure R16), attributed to the electron-withdrawing effects of F/O atoms in MTFSI, which reduce the electron density of the EA molecules. Concurrently, MTFSI signals shift upfield with increasing EA content (Figure R17), confirming the presence of non-covalent interactions between EA and MTFSI. Furthermore, this type of polymer-anion interaction is well-documented in dynamic crosslinking for functional polymers. For instance, Cao et al. used this interaction between poly(ethyl acrylate) backbone and TFSI[–] anion to prepare tough ionogels, with the –CH₂–CH₃⋯CF₃ interaction confirmed by FTIR [ACS Appl. Polym. Mater., 2020, 2, 2359–2365]. Similarly, Liu’s group demonstrated the use of this interaction in synthesizing robust and stretchable elastomers, with evidence provided by both NMR and FTIR techniques [Materials Horizons, 2020, 7(3): 912-918]. We have revised the manuscript accordingly on page 2, lines 23 to 33, to reflect these additional details. Thank you again for your valuable input.

“As evidence, the magic angle spinning solid-state NMR (MAS-ssNMR) spectra presented in Figure 1b and Figure S1 demonstrate that as the (trifluoromethane) sulfonimide lithium methacrylate (MTFSI) content increases, the signals of ethylene acrylate (EA) shift gradually downfield. This shift can be attributed to the electron-withdrawing effects of the F/O atoms in MTFSI. Conversely, with increasing EA content, the signals of MTFSI shift upfield (Figure S1c), which further supports the presence of non-covalent interactions between EA and MTFSI.

... Consequently, this non-covalent interaction is extensively utilized in developing functional polymers, including mechanically robust ionogels²³ and stretchable elastomers²⁴. In this work,..."

Figure R15 (Figure S1a). Full range Magic Angle Spinning Solid-State NMR (MAS-ssNMR) spectra of EA/MTFSI mixture at different mass ratio. The spectra were referenced against trimethylsilane (TMS, 0 ppm). The spinning rate of sample is 2000 Hz.

Figure R16 (Figure 1b and Figure S1b). Enlarged MAS-ssNMR spectra showing the downfield shifting of EA signals with increasing MTFTSI content.

Figure R17 (Figure S1c). Enlarged MAS-ssNMR spectra showing the upfield shifting of MTFSI signals with increasing EA content.

3. As described in Fig. 1a, the authors attribute the self-healing ability of plastic ceramic electrolyte to the non-covalent interaction between $-\text{CH}_3\cdots\text{CF}_3$. Such interactions include mainly those. If only weak interaction forces, such as van der Waals forces, are sufficient to support their self-healing.

Reply: We appreciate your insightful comments. In response to Q2, we have demonstrated the existence of $-\text{CH}_2-\text{CH}_3\cdots\text{CF}_3$ interaction using concentration-dependent MAS-ssNMR techniques. Regarding the strength of these non-covalent interactions, previous theoretical predictions indicate a high binding energy of 0.4-0.5 eV. For instance, Jung et al. employed DFT calculations and reported a binding energy of -0.457 eV between TFSI^- and the $-\text{CH}_2-$ group in a polyester backbone [Adv. Mater. 2018, 30, 1706851]. Similarly, Wang et al. used molecular dynamics simulations and reported a binding energy of -14.6 kcal/mol (-0.62 eV) between the FSI^- anion and the $-\text{CH}_2-$ group in the PEO backbone [Nature Materials, 21(9), 1057-1065], which is consistent with Jung's findings. In contrast, the strength of water-water hydrogen bonding is reported to be -0.25 eV [Physical Review B, 2006, 74(24): 245409], which is lower than the binding energies of the above non-covalent interactions discussed. Therefore, we believe that the $-\text{CH}_2-\text{CH}_3\cdots\text{CF}_3$ interaction is sufficiently robust to support the self-healing capability of our plastic ceramic electrolyte. We are grateful for your constructive feedback, which has enhanced our understanding of polymer-anion non-covalent interactions. We have revised our manuscript accordingly on page 2, lines 29 to 31:

“...which supports the presence of non-covalent interactions between EA and MTFSI. **Previous theoretical predictions also show a high binding energy of 0.4-0.5 eV for this $-\text{CH}_3\cdots\text{CF}_3$ interaction^{20,21}, which surpasses the strength of water-water hydrogen bonding (0.25 eV)²².**”

4. As described in Fig. 2a and Fig. 2c, in the operando XRF images, the sulfur element in SPE (highlighted in green) and the phosphorus element in LATP (highlighted in red) are unevenly distributed, does this uneven distribution affect the homogeneity of the use.

Reply: Thank you for your careful observation and professional suggestions. The uneven distribution of SPE and LATP is due to the infiltration of SH-SPE and the diffusion of LATP within the polymer network. Since this uneven distribution was observed at the micrometer scale, we believe it does not cause macroscopic-scale phase separation. This is evidenced by the uniform macroscopic compatibility between SH-SPE and LATP shown in Figure 1c. Therefore, the observed micrometer-scale inhomogeneity in the XRF images will not affect the electrolyte fabrication or battery assembly processes.

Regarding the effect on dendrite-inhibiting capability, this micrometer-scale inhomogeneity could possibly result in local regions with low LATP concentration, corresponding lower modulus, and inferior dendrite inhibition ability. As shown in Figure 2a, this inhomogeneity was observed after 10 hours of battery operation. This suggests that the SH-SPE infiltration and LATP diffusion is a dynamic process, meaning the concentration of LATP and corresponding local modulus could change over time. Additionally, the varying content of LATP in different regions may create a concentration gradient, promoting the diffusion of LATP to low modulus regions and the infiltration of SH-SPE to high modulus regions. Therefore, with extended battery cycling, we think this inhomogeneity may gradually decrease. Consequently, we believe this uneven distribution of LATP and SH-SPE will not significantly affect the dendrite inhibition capability of PCE, as evidenced by the excellent durability of the $\text{Li}^0/\text{PCE}/\text{Li}^0$ cell, which demonstrates a cycling life of 2000 hours at $1 \text{ mA}/\text{cm}^2$ and 4200 hours at $0.2 \text{ mA}/\text{cm}^2$ (Figure 1e and Figure 5a). Thanks again for your valuable input and we have revised the manuscript correspondingly on Page 5, lines 25 to 32:

“...at $1 \text{ mA}/\text{cm}^2$ ($4.82 \text{ }\mu\text{m}/\text{hour}$). The observed uneven distribution of LATP and SH-SPE resulted from the infiltration of SH-SPE and the diffusion of LATP, indicating that the SH-SPE infiltration and LATP diffusion is a dynamic process. Additionally, the varying LATP content in different regions could create a concentration gradient, promoting the diffusion of LATP to low concentration regions. Therefore, with extended battery cycling, we believe this inhomogeneity may gradually decrease and will not significantly affect the dendrite inhibition capability of PCE, as evidenced by the excellent durability of the $\text{Li}^0/\text{PCE}/\text{Li}^0$ cell.”

5. As described in Fig. 2c and Fig. 2d, the authors find that the self-healing process significantly accelerated with decreased void size. What caused this.

Reply: Thank you for your professional question. The self-healing process described in this study involves the infiltration of the polymer followed by the migration of the ceramic phase to fill the voids. Since voids are three-dimensional, the volume of a void is proportional to the cube of its diameter. For instance, repairing a $200\text{-}\mu\text{m}$ sized void requires approximately eight times more PCE infiltration compared to repairing a $100\text{-}\mu\text{m}$ sized void. Consequently, the self-healing process is significantly accelerated with decreased void size due to the much smaller volume and, thus, the reduced amount of material required for self-healing. We have revised the manuscript accordingly on page 5, lines 36-39 to incorporate your thoughtful inputs:

“...which is much faster than the Li^0 -deposition speed at $1 \text{ mA}/\text{cm}^2$ ($4.82 \text{ }\mu\text{m}/\text{hour}$). The accelerated self-healing rate can be attributed to the three-dimensional nature of voids, where the volume of a void

is proportional to the cube of its diameter. Consequently, the self-healing process is significantly accelerated at smaller void sizes due to the much smaller volume.”

6. As described in Fig. 2c, the two 300- μm -sized voids were completely self-repaired within 20 hours. Are there advantages over other self-healing classes of solid electrolytes. The self-healing process requires in 0.5 C and 20 hours cycling, whether it is feasible in practical applications.

Reply: Thank you for your constructive suggestions. Firstly, previous self-healing systems generally rely on protic moieties such as $-\text{OH}$ and $-\text{NH}$ [J. Am. Chem. Soc. 2019, 141, 18932–18937; ACS Macro Lett. 2020, 9, 525–532], which are highly reactive with the Li^0 anode. In contrast, we employed aprotic $-\text{CH}_2-\text{CH}_3$ moieties in this study, which enable better interfacial stability with the Li^0 anode. Secondly, prior studies have only demonstrated self-healing capabilities *ex-situ*, without quantifying the self-healing rate. To the best of our knowledge, this is the first study to reveal and quantify the real-time self-healing mechanism during battery cycling, demonstrating a fast self-healing rate of 22.6 $\mu\text{m}/\text{hour}$. Thirdly, our PCE exhibits a dual-phase self-healing process, allowing ceramic particles to migrate through the polymer network and fill voids. The high modulus nature of the ceramic particles could also contribute to dendrite inhibition. These advantages have led to greatly enhanced cycling durability of our PCE compared with previous self-healing solid electrolytes (Table R2).

In this study, XRF results show that the PCE was able to repair 300- μm -sized voids within 20 hours of cycling at 0.2 mA/cm^2 . In most practical applications, such as electric vehicles and cellphones, batteries might be charged once every 24 hours and discharged over a period of 10-50 hours. Additionally, considering the much smaller size of Li^0 dendrites (100-200 nm in diameter and 2-4 μm in length, [Nature nanotechnology, 2019, 14(11): 1042-1047]) as well as the accelerated self-healing rate of PCE at smaller void sizes, the required time for repairing dendrite-induced voids would be much shorter than 20 hours. Therefore, we believe the typical usage patterns and charging cycles in practical applications match the self-healing rate of our PCE. Correspondingly, we have revised the manuscript on page 17, lines 39-47 to incorporate your thoughtful inputs, thank you.

“Compared with previous self-healing solid electrolytes, this study employed aprotic $-\text{CH}_2-\text{CH}_3$ moieties, avoiding the use of Li^0 -reactive $-\text{OH}$ and $-\text{NH}$ moieties^{26,27}, which enables better interfacial stability with the Li^0 anode. Additionally, this study reveals and quantifies the real-time self-healing mechanism during battery cycling, demonstrating a fast self-healing rate of 22.6 $\mu\text{m}/\text{hour}$. Furthermore, the PCE exhibits a dual-phase self-healing process, allowing ceramic particles to migrate through the polymer network and fill voids. The high modulus of the ceramic particles could also contribute to dendrite inhibition. These advantages have led to significantly enhanced cycling durability compared with previous self-healing solid electrolytes (Table S3). In practical applications, batteries might be charged once every 24 hours and discharged over a period of 10-50 hours. These typical usage patterns and charging cycles align well with the self-healing rate of our PCE.”

Table R2 (Table S3). Comparison of the Li^0 - Li^0 symmetric cell durability and full cell performance with previous Li^0 -anode batteries on self-healing solid polymer electrolytes.

a) Critical current density; b) 3V-class cathode: LiFePO_4 ; 4V-class cathode: NMC, LiCoO_2 , etc.

References	CCD ^{a)} (mA/cm ²)	Li ⁰ -Li ⁰ cell Current/life/temp.	Current× cycle life (mAh/cm ²)	Cathode ^{b)}	Current Density	capacity/cycle life/retention	Loading (mg/cm ²)	Loading (mAh/cm ²) ×cycle number
This work	30	1/2900hr/r.t.	2900	3 V	C/2	127/2400/84%	4.5	1836
		2/1000hr/50°C	2000	3 V	2C	123/4000/88%	2.3	1564
		5/300hr/50°C	1500	4 V	1C	144/2860/80%	3	1844
		10/150hr/50°C	1500	4V	C/6	130/1000/71%	7.4	1600
2024 Nat. Commun. ³³	NA	0.2/6000hr/60°C	300	3 V (SPAN)	0.3C	602/400/99.7%	2.1	506
2024 Adv. Funct. Mater. ³⁴	0.5	0.2/2100hr/NA	105	3 V	0.3C	157/480/80%	6	490
				4 V	0.3C	167/330/85%	1.5	106
2023 Angew. Chem. ¹⁰	0.3	0.1/1000hr/60°C	100	3 V	0.2C	149/600/75%	1.2	122
2023 Mater. Horiz. ³⁵	NA	0.05/1600hr/r.t.	80	3 V	0.5C	134/400/68%	2.8	190
2023 ACS Appl. Energy Mater. ³⁶	1	1/20hr/25°C	20	NA	NA	NA	NA	NA
2023 Materials Today Energy ³⁷	NA	0.05/1400hr/r.t.	70	3 V	0.2C	150/300/94%	1.4	71
2023 Adv. Funct. Mater. ³⁸	NA	0.15/1300hr/NA	195	3 V	0.5C	151/230/98%	3	117
				4 V	0.3C	186/120/80%	3	77
2022 Energy Storage Materials ³⁹	0.1	0.05/500hr/60°C	25	3 V	0.1C	160/100/92%	NA	NA
2022 Nano Energy ⁴⁰	NA	0.5/1200hr/NA	600	4 V	1C	128/300/99%	4.6	297
2020 Angew. Chem. ⁴¹	NA	0.2/600hr/NA	120	4 V	0.1C	115/200/86%	1.5	64
2020 ACS Energy Lett. ⁴²	NA	0.1/700hr/NA	70	4 V	0.1C	112/120/80%	1.2	31
2020 ACS Macro Lett. ⁴³	0.1	0.05/200hr/60°C	10	3 V	0.1C	145/70/79%	NA	NA
2019 J. Am. Chem. Soc. ⁴⁴	NA	0.015/0.44hr/60°C	0.006	NA	NA	NA	NA	NA

7. As described in Fig. 4, the authors find that both the polymer phase and ceramic phase serve as effective ion conduction pathways. However, due to the difference in ionic conductivity between the polymer phase and the ceramic phase, does it affect the transport of lithium ions, leading to non-uniform deposition and the creation of dendrites.

Reply: Thank you for your insightful question. We agree that differences in ionic conductivity could potentially cause lithium ions to preferentially transport through the more conductive phase, resulting in higher local current density and leading to non-uniform lithium deposition [Journal of Energy Chemistry 85 (2023) 181–190]. In this study, the pristine SH-SPE exhibits a room temperature conductivity of 1.1 mS/cm (Figure R18), while the pristine LATP has a lower room temperature conductivity of 0.024 mS/cm due to its high grain boundary resistance. However, the infiltration of SH-SPE into the grain boundaries of LATP (Figure 1d) significantly enhances Li⁺ conduction between LATP particles, reducing the grain boundary resistance to 55 ohm·cm², corresponding to a grain boundary conductivity of 0.8 mS/cm. Thus, in the PCE, both the polymer phase and ceramic phase show comparable conductivities of 1.1 mS/cm and 0.8 mS/cm, respectively.

Additionally, we employed solid-state NMR and a ⁷Li to ⁶Li isotope exchange approach to determine the preferred route for Li⁺ conduction. As shown in the ⁶Li NMR spectra (Figure R19), after cycling in a ⁶Li/PCE/⁶Li symmetric cell, the integral area for the SH-SPE signal increased by 7.1-fold, while

the integral area for the LATP signal increased by 6.3-fold, which is close to that of SH-SPE. This result suggests that Li^+ transport equally through both the polymer and ceramic phases, thereby eliminating the non-uniform deposition and dendrite formation that could arise from differences in ionic conductivity between the phases. Thanks again for your valuable input. We have revised the manuscript accordingly on page 10, lines 9 to 14.

“after isotope exchange from ^7Li to ^6Li , the integral area for the SH-SPE signal increased by 7.1-fold, while the integral area for the LATP signal increased by 6.3-fold, which is comparable to that of SH-SPE. This result suggests that Li^+ ions transport equally through both the polymer and ceramic phases, due to their comparable conductivities (Figure S36). This uniform transport mitigates the uneven deposition and dendrite formation that could arise from differences in ionic conductivity between the phases³⁶”

Figure R18 (Figure S36). **a**, EIS plot of SS/SH-SPE/SS showing ionic conductivity of SH-SPE. SS refers to stainless steel blocking electrodes. **b**, EIS plots of SS/LATP/SS and SS/PCE/SS cells showing the grain boundary resistance and ionic conductivity.

Note: The differences in ionic conductivity between phases could cause Li^+ to preferentially transport through the more conductive phase, resulting in higher local current density and leading to non-uniform lithium deposition. In this study, the pristine SH-SPE exhibits a room temperature conductivity of 1.1 mS/cm, while the pristine LATP has a lower room temperature conductivity of 0.024 mS/cm due to its high grain boundary resistance. However, the infiltration of SH-SPE into the grain boundaries of LATP (Figure 1d) significantly enhances Li^+ conduction between LATP particles, reducing the grain boundary resistance to $55 \text{ ohm}\cdot\text{cm}^2$, corresponding to a grain boundary conductivity of 0.8 mS/cm. Thus, in the PCE, both the polymer phase and ceramic phase show comparable conductivities of 1.1 mS/cm and 0.8 mS/cm, respectively.

Figure R19 (Figure 4d-4e). **a**, Schematic illustration of the isotope exchange method for revealing the ion conduction pathway. **b**, ^6Li solid-state NMR spectra of the pristine PCE and the PCE cycled in ^6Li - ^6Li symmetric cells.

8. As described in Fig. 6f and 6g, what might be the reason for the lower efficiency of the battery during the first 5 to 10 laps, this phenomenon does not seem to occur in the LFP battery system.

Reply: Thank you for your insightful question. The lower coulombic efficiency observed during the first 5-10 cycles may be attributed to side reactions between the PCE and the cathode at high voltage (4.2 V). While LATP is known for its good electrochemical stability up to 4.2 V [J. Mater. Chem. A, 2016, 4, 3253], polymer electrolytes often face challenges with insufficient oxidation resistance when coupled with high-voltage cathodes. In contrast, the LFP-based battery operates at a lower cut-off voltage of 3.8 V, thereby reducing the risk of polymer electrolyte oxidation and associated side reactions. To address this issue in future studies, the incorporation of functional additives, such as lithium difluoro(oxalato)borate (LiDFBOB), could be considered. These additives can facilitate the formation of a stable cathode-electrolyte interface (CEI), which is anticipated to enhance electrochemical stability and mitigate side reactions [Nano Energy, 72 (2020), 104655]. While this study primarily focuses on investigating and characterizing the self-healing capability of the PCE to enhance cycling durability with the Li^0 anode, optimization of electrolyte-cathode interfaces will be a key area of exploration in following research. We appreciate your valuable suggestions and have incorporated your thoughtful inputs on page 14, lines 31-37 of the manuscript:

“Despite demonstrating good cycling durability, the initial low coulombic efficiency and discharge capacity need further improvement. These issues may stem from potential side reactions at the cathode-electrolyte interface and high resistance at room temperature, which contributes to voltage hysteresis⁴⁰. Future studies will focus on enhancing the initial coulombic efficiency by developing a stable cathode-electrolyte interface (CEI)⁴¹ and addressing the low discharge capacity through the design of composite cathodes to improve ion conduction within the electrode.”

9. As described in Fig. 6g, the H-SSE was paired with a high-Ni, zero-Co, zero-strain cathode in coin cells. The zero-strain test is more meaningful in pouch cells than in coin cells. What happens in pouch cells.

Reply: Thank you for your insightful comments and suggestions. In this revised manuscript, we have evaluated the performance of H-SSE when paired with the zero-strain cathode in pouch cell (Figure R20). The effective area of the zero-strain cathode is 7.8 cm^2 , with a mass loading of 3.06 mg/cm^2 . Although no external pressure was applied, the pouch cell demonstrated good durability for 400 cycles

when operating at $C/2$ and $50\text{ }^{\circ}\text{C}$. The initial discharge capacity was 141.8 mAh/g , and the capacity retention after 400 cycles was 79% . The average coulombic efficiency from the 1st to the 400th cycle was 99.94% . We have revised the manuscript to incorporate these results on page 14, lines 46-48.

“...indicating the critical role of PCE as a dendrite inhibiting layer to improve full cell durability. To further evaluate the potential of the PCE in practical applications, we tested the performance of H-SSE when paired with the zero-strain cathode in pouch cells (Figure S37). Despite the absence of external pressure, the pouch cell demonstrated an initial discharge capacity of 141.8 mAh/g at $C/2$ and $50\text{ }^{\circ}\text{C}$, maintaining good durability over 400 cycles. The capacity retention was 79% after 400 cycles, and the average coulombic efficiency from the 1st to the 400th cycle was 99.94% .”

Figure R20 (Figure S37). **a**, Performance of H-SSE when paired with a zero-strain cathode in pouch cells. The effective area of the zero-strain cathode is 7.8 cm^2 , with a mass loading of 3.06 mg/cm^2 . The cell was cycled at $C/2$ and $50\text{ }^{\circ}\text{C}$, with cut-off voltage of $2.8\text{-}4.3\text{ V}$. **b**, Charge-discharge profiles of pouch cell when cycling at $C/2$ and $50\text{ }^{\circ}\text{C}$.

10. In the synthesis of the plastic ceramic electrolyte, a large number of polymer electrolytes are used. What are the advantages of this composite plastic ceramic electrolyte over traditional polymer electrolytes.

Reply: We appreciate your constructive suggestions. Compared with traditional polymer electrolytes, this study reports a composite electrolyte with high ceramic content ($70\text{ wt}\%$) that combines the advantages of both polymer and ceramic electrolytes. The high modulus of ceramic particle provides a physical barrier to block dendrite penetration, thereby extending the cycling life of Li^0 -anode batteries. As shown in Figure 5a, when cycling at 0.2 mA/cm^2 and 0.5 mAh/cm^2 , the pristine SH-SPE exhibited cycling durability of ~ 1000 hours. After introducing $70\text{ wt}\%$ of LATP ceramic, the cycling life was dramatically improved to 4200 hours. Additionally, the single Li^+ conducting nature of LATP ceramic improves the Li^+ transfer number (t_{Li^+}). The PCE exhibits a high t_{Li^+} of 0.74 (Figure S11), which is much higher than that of conventional polymer electrolytes (e.g., the t_{Li^+} of conventional PEO-SPE is $0.2\text{-}0.4$). This high t_{Li^+} enhances effective Li^+ conductivity and minimizes dendrite formation by reducing concentration polarization [Journal of The Electrochemical Society, 161 (6) A847-A855 (2014)]. More importantly, this PCE demonstrates a unique dual-phase self-healing mechanism with a fast healing rate of $22.6\text{ }\mu\text{m/hour}$, effectively eliminating "hot spots" for dendrite formation, such as voids and cracks. Attributing to above benefits, the PCE has enabled notable improvements in Li^0 anode compatibility, durability, and areal capacity compared to traditional polymer electrolytes (Table R3 and R4), thereby affirming the effectiveness of our design in achieving durable and high-performance solid-state batteries. Once again, we appreciate your constructive

suggestions, and have revised the manuscript on page 17, lines 31 to 38 to incorporate your valuable inputs:

“Compared with traditional polymer electrolytes, the PCE combines the advantages of both polymer and ceramic electrolytes. The high modulus of ceramic particles provides a physical barrier to block dendrite penetration, thereby extending the cycling life of Li⁰-anode batteries. Additionally, the single Li⁺ conducting nature of LATP ceramic enables a high Li⁺ transfer number (t_{Li^+}) of 0.74 (Figure S11), which enhances effective Li⁺ conductivity and minimizes dendrite formation by reducing concentration polarization⁴⁸. More importantly, this PCE demonstrates a unique dual-phase self-healing mechanism with a fast healing rate of 22.6 $\mu\text{m}/\text{hour}$, effectively eliminating “hot spots” for dendrite formation, such as voids and cracks.”

Table R3 (Table S1). Comparison of the Li⁰-Li⁰ symmetric cell durability and full cell performance with previous Li⁰-anode batteries on PEO-based solid polymer electrolytes.

a) Critical current density; b) 3V-class cathode: LiFePO₄; 4V-class cathode: NMC, LiCoO₂, etc.

References	CCD ^{a)} (mA/cm ²)	Li ⁰ -Li ⁰ cell Current/life/temp.	Current \times cycle life (mAh/cm ²)	Cathode ^{b)}	Current Density	capacity/cycle life/retention	Loading (mg/cm ²)	Loading (mAh/cm ²) \times cycle number
This work	30	1/2900hr/r.t.	2900	3 V	C/2	127/2400/84%	4.5	1836
		2/1000hr/50°C	2000	3 V	2C	123/4000/88%	2.3	1564
		5/300hr/50°C	1500	4 V	1C	144/2860/80%	3	1844
		10/150hr/50°C	1500	4V	C/6	130/1000/71%	7.4	1600
2024 Nat. Energy ¹	3.7	0.5/1000hr/	500	4V	C/2	210/400/80%	5.3	460
2024 Nat. Nanotechnol. ²	NA	0.1/1000hr/60°C	100	3 V	0.5C	138/200/90%	1.5	51
				3 V	0.07C	130/35/95%	6.8	41
2024 Nat. Commun. ³	0.5	0.1/2250hr/90°C	225	3 V	0.5C	160/1000/78%	1.6	270
2024 Nat. Mater. ⁴	NA	0.05/1500hr/30°C	75	3 V	0.05C	147/40/92%	8.8	53
2024 Angew. Chem. ⁵	1.1	0.4/1500hr/60°C	600	3V	2C	123/1500/80%	2	510
2024 Energy Environ. Sci. ⁶	NA	0.1/100hr/25°C	10	3 V	0.1C	147/100/88%	1.2	21
				4 V	0.1C	152/100/66%	1.2	26
2024 Nano Energy ⁷	0.5	0.1/1600hr/NA	160	4 V	0.1C	166/250/92%	3	161
		0.2/400hr/NA	80	4 V	NA	178/100/82%	10	215
2024 ACS Energy Lett. ⁸	NA	0.4/500hr/70°C	200	3 V	2C	130/1000/76%	2.5	425
2023 J. Am. Chem. Soc. ⁹	NA	0.1/400hr/25°C	40	3 V	0.5C	130/250/96%	2.5	106
				4 V	0.1C	134/100/84%	2.5	42.5
2023 Angew. Chem. ¹⁰	0.3	0.1/1000hr/60°C	100	3 V	0.2C	149/600/75%	1.2	122
2023 PNAS ¹¹	2	0.5/700hr/25°C	350	3 V	0.5C	135/300/92%	2.5	117
				3 V	0.2C	160/100/80%	8	136
2023 Nano Energy ¹²	NA	0.1/2000hr/60°C	200	3 V	1C	127/300/99%	1	51
2023 Nano Energy ¹³	NA	0.5/600hr/NA	300	4 V	0.5C	145/100/92%	2.2	28
2023 Nano Energy ¹⁴	0.7	0.1/900hr/25°C	90	3 V	0.2C	155/330/95%	1.7	95

				4 V	0.1C	152/100/83%	1.1	24
2023 Nano Energy ¹⁵	0.2	0.1/320hr/60°C	32	3 V	0.1C	148/100/93%	NA	NA
2023 Adv. Energy Mater. ¹⁶	NA	0.3/500hr/60°C	150	3 V	1C	147/300/99.7%	2	102
2022 Joule ¹⁷	NA	0.75/1300hr/r.t.	974	3 V	0.25C	160/100/97%	4	68
2022 ACS Energy Lett. ¹⁸	NA	0.1/445hr/70°C	45	4 V	C/20	160/80/80%	4.7	80
2022 Nano Energy ¹⁹	NA	0.2/2400hr/60°C	480	3 V	0.5C	133/1350/83%	3.9	895
2022 Adv. Energy Mater. ²⁰	0.9	0.5/300hr/60°C	150	3 V	0.5C	160/500/70%	3	255
2022 Energy Environ. Sci. ²¹	NA	0.2/2200hr/NA	440	4 V	0.5C	188/1000/80%	2.5	537
2022 Nat. Commun. ²²	NA	0.1/3800hr/90°C	380	3 V	0.5C	158/400/98%	1.5	102
				4 V	0.5C	145/100/87%	1.5	25.2
2022 Nat. Commun. ²³	1.2	0.1/1200hr/NA	120	3 V	0.25C	167/1200/87%	2	480
		1.2/120hr/NA	144					
2022 J. Am. Chem. Soc. ²⁴	NA	0.1/500hr/70°C	50	3 V	C/3	151/200/86%	4	136
2021 J. Am. Chem. Soc. ²⁵	2	0.4/400hr/NA	160	3 V	0.15C	160/120/78%	5	102
		1.2/100hr/NA	120	4 V	0.24C	140/120/86%	3	77.4
2021 Adv. Mater. ²⁶	NA	0.1/1200hr/NA	120	4 V	0.1C	177/100/78%	21	450
		1/1100hr/NA	1100	4 V	0.1C	163/100/89%	23	450
2021 Nano Lett. ²⁷	NA	0.1/300hr/60°C	30	3 V	1C	146/1000/84%	1	170
2020 Adv. Mater. ²⁸	0.5	0.1/1300hr/NA	130	3 V	0.5C	144/500/86%	2.4	204
2020 Energy Environ. Sci. ²⁹	NA	0.2/2500hr/NA	500	3 V	C/3	160/210/97%	3.3	118
				4 V	0.1C	135/110/90%	3.5	82
2020 Nano Lett. ³⁰	NA	0.1/300hr/60°C	30	3 V	0.2C	140/300/95%	1.5	76.5
2020 Nano Lett. ³¹	NA	0.05/600hr/25°C	30	3 V	0.1C	166/250/77%	5	212
2020 Nano Lett. ³²	NA	1/560hr/90°C	560	3 V	1C	140/200/86%	2.5	85
		2/260hr/90°C	520	4 V	C/20	160/50/80%	2.5	27

Table R4 (Table S2). Comparison of the Li⁰-Li⁰ symmetric cell durability and full cell performance with previous Li⁰-anode batteries on beyond-PEO solid polymer electrolytes.

a) Critical current density; b) 3V-class cathode: LiFePO₄; 4V-class cathode: NMC, LiCoO₂, etc.

References	CCD ^{a)} (mA/cm ²)	Li ⁰ -Li ⁰ cell Current/life/temp.	Current× cycle life (mAh/cm ²)	Cathode ^{b)}	Current Density	capacity/cycle life/retention	Loading (mg/cm ²)	Loading (mAh/cm ²) ×cycle number
This work	30	1/2900hr/r.t.	2900	3 V	C/2	127/2400/84%	4.5	1836
		2/1000hr/50°C	2000	3 V	2C	123/4000/88%	2.3	1564
		5/300hr/50°C	1500	4 V	1C	144/2860/80%	3	1844
		10/150hr/50°C	1500	4 V	C/6	130/1000/71%	7.4	1600
2024 Nano Lett. ⁴⁵	0.5	0.1/1500hr/60°C	150	3 V	0.2C	160/300/99%	1.9	97
		0.5/400hr/60°C	200	4 V	0.2C	150/1/NA	1.9	0.4
2024 Angew. Chem. ⁴⁶	1	0.1/200hr/NA	20	3 V	0.5C	140/500/70%	1.5	128
				4 V	0.2C	271/15/99%	11	45

2023 Nat. Commun. ⁴⁷	NA	0.1/2500hr/NA	250	4 V	0.5C	164/200/89%	0.8	34.4
2023 Angew. Chem. ⁴⁸	NA	0.1/500hr/60°C	50	3 V	0.1C	150/440/99%	2.5	187
2023 Angew. Chem. ⁴⁹	NA	0.1/1000hr/60°C	100	4 V	0.1C	175/200/70%	3	129
2023 Angew. Chem. ⁵⁰	NA	0.1/1400hr/60°C	140	3 V	1C	137/350/97%	2	119
		0.2/800hr/60°C	160	3 V	2C	114/700/80%	2	238
2023 Adv. Mater. ⁵¹	NA	0.5/700hr/NA	350	4 V	0.5C	170/700/84%	1.8	270
2023 Adv. Mater. ⁵²	NA	1/1000hr/NA	1000	3 V	2C	128/600/92%	1.2	122
2023 ACS Energy Lett. ⁵³	0.5	0.05/2000hr/40°C	100	4 V	2C	140/300/84%	2.3	148
				4 V	0.2C	160/100/86%	10.6	228
2022 Nature ⁵⁴	NA	10/1500hr/r.t.	15000	3 V	1C	93/1000/95%	1.5	260
				4 V	0.3C	110/100/88%	10	213
2022 Angew. Chem. ⁵⁵	NA	0.5/300hr/45°C	150	3 V	0.15C	159/350/82%	3.2	190
2022 Adv. Mater. ⁵⁶	2.4	0.1/4000hr/NA	400	3 V	0.4C	160/300/96%	1.5	76.5
2022 Energy Environ. Sci. ⁵⁷	NA	0.5/2200hr/NA	1100	3 V	0.5C	122/1200/99%	1	204
		1/800hr/NA	800	4 V	2C	160/200/82%	4	172
2022 Nano Energy ⁵⁸	NA	0.5/800hr/NA	400	4 V	1C	150/100/89%	2	32
2022 Nano Energy ⁵⁹	2	0.1/2200hr/NA	220	4 V	2C	133/1500/74%	2	645
2022 Nano Energy ⁶⁰	2.4	0.2/3000hr/NA	600	3 V	2C	140/200/67%	2	68
				4 V	0.2C	120/150/75%	4	129
2022 Nano Energy ⁴⁰	BA	0.5/1200hr/NA	600	4 V	1C	128/300/99%	4.6	297
2022 Adv. Energy Mater. ⁶¹	0.5	0.3/550hr/NA	165	4 V	2C	130/300/90%	2.3	148
2021 Nat. Mater. ⁶²	1	0.2/2000hr/r.t.	400	NA	NA	NA	NA	NA
2020 Adv. Energy Mater. ⁶³	NA	0.1/800hr/NA	80	4 V	0.5C	191/200/85%	1.5	64

Reviewer #4 (Remarks to the Author):

Reply: Thank you for your thorough review. We greatly appreciate your insightful suggestions, which have significantly contributed to the improvement of our manuscript.

Reviewer #5 (Remarks to the Author):

Reply: Thank you for your thorough review. We greatly appreciate your insightful suggestions, which have significantly contributed to the improvement of our manuscript.

References

1. Zhang, W. *et al.* Single-phase local-high-concentration solid polymer electrolytes for lithium-metal batteries. *Nature Energy* **9**, 386-400 (2024).
2. Wan, J. *et al.* Ultrathin, flexible, solid polymer composite electrolyte enabled with aligned nanoporous host for lithium batteries. *Nat Nanotechnol* **14**, 705-711 (2019).
3. Zhu, G. R. *et al.* Non-flammable solvent-free liquid polymer electrolyte for lithium metal batteries. *Nat Commun* **14**, 4617 (2023).
4. Han, S. *et al.* Sequencing polymers to enable solid-state lithium batteries. *Nat Mater* **22**, 1515-1522 (2023).
5. Cheng, Y. *et al.* Zwitterionic Cellulose-Based Polymer Electrolyte Enabled by Aqueous Solution Casting for High-Performance Solid-State Batteries. *Angew Chem Int Ed Engl* **63**, e202400477 (2024).
6. Li, R. *et al.* The deconstruction of a polymeric solvation cage: a critical promotion strategy for PEO-based all-solid polymer electrolytes. *Energy & Environmental Science* **17**, 5601-5612 (2024).
7. Gong, Y. *et al.* Ultra-thin and high-voltage-stable Bi-phasic solid polymer electrolytes for high-energy-density Li metal batteries. *Nano Energy* **119** (2024).
8. Guo, K. *et al.* In Situ Orthogonal Polymerization for Constructing Fast-Charging and Long-Lifespan Li Metal Batteries with Topological Copolymer Electrolytes. *ACS Energy Letters* **9**, 843-852 (2024).
9. Ding, P. *et al.* Molecular Self-Assembled Ether-Based Polyrotaxane Solid Electrolyte for Lithium Metal Batteries. *J Am Chem Soc* **145**, 1548-1556 (2023).
10. Chen, J. *et al.* Multiple Dynamic Bonds-Driven Integrated Cathode/Polymer Electrolyte for Stable All-Solid-State Lithium Metal Batteries. *Angew Chem Int Ed Engl* **62**, e202307255 (2023).
11. Li, C. *et al.* An intrinsic polymer electrolyte via in situ cross-linked for solid lithium-based batteries with high performance. *PNAS Nexus* **2**, pgad263 (2023).
12. Han, L. *et al.* Noncombustible 7 μm -thick solid polymer electrolyte for highly energy density solid state lithium batteries. *Nano Energy* **112** (2023).
13. Saleem, A. *et al.* Boosting lithium-ion conductivity of polymer electrolyte by selective introduction of covalent organic frameworks for safe lithium metal batteries. *Nano Energy* **128** (2024).
14. Zheng, J. *et al.* Heterocyclic polymer supported cathode/Li interface layers to lower the operational temperature of PEO-based Li-batteries. *Nano Energy* **118** (2023).
15. Chen, Z., Jia, H., Yan, S. & Gohy, J.-F. Polymer-coated silica dual functional fillers to improve the performance of poly(ethylene oxide)-based solid electrolytes. *Nano Energy* **114** (2023).
16. Kim, E. *et al.* Functionality of 1-Butyl-2,3-Dimethylimidazolium Bromide (BMI-Br) as a Solid Plasticizer in PEO-Based Polymer Electrolyte for Highly Reliable Lithium Metal Batteries. *Advanced Energy Materials* **13** (2023).
17. Cheng, Q. *et al.* Stabilizing lithium plating in polymer electrolytes by concentration-polarization-induced phase transformation. *Joule* **6**, 2372-2389 (2022).
18. Arrese-Igor, M. *et al.* Toward High-Voltage Solid-State Li-Metal Batteries with Double-Layer Polymer Electrolytes. *ACS Energy Letters* **7**, 1473-1480 (2022).
19. Yang, L. *et al.* The plasticizer-free composite block copolymer electrolytes for ultralong lifespan all-solid-state lithium-metal batteries. *Nano Energy* **100** (2022).
20. Ma, Y. *et al.* Scalable, Ultrathin, and High-Temperature-Resistant Solid Polymer Electrolytes for Energy-Dense Lithium Metal Batteries. *Advanced Energy Materials* **12** (2022).
21. Wang, H. *et al.* A strongly complexed solid polymer electrolyte enables a stable solid state high-voltage lithium metal battery. *Energy & Environmental Science* **15**, 5149-5158 (2022).
22. Su, Y. *et al.* Rational design of a topological polymeric solid electrolyte for high-performance all-solid-state alkali metal batteries. *Nat Commun* **13**, 4181 (2022).
23. Hu, J. *et al.* Dual fluorination of polymer electrolyte and conversion-type cathode for high-capacity all-solid-state lithium metal batteries. *Nat Commun* **13**, 7914 (2022).

24. Qiao, L. *et al.* Anion pi-pi Stacking for Improved Lithium Transport in Polymer Electrolytes. *J Am Chem Soc* **144**, 9806-9816 (2022).
25. Xu, B. *et al.* Interfacial Chemistry Enables Stable Cycling of All-Solid-State Li Metal Batteries at High Current Densities. *J Am Chem Soc* **143**, 6542-6550 (2021).
26. He, F., Tang, W., Zhang, X., Deng, L. & Luo, J. High Energy Density Solid State Lithium Metal Batteries Enabled by Sub-5 microm Solid Polymer Electrolytes. *Adv Mater* **33**, e2105329 (2021).
27. Han, L. *et al.* Flame-Retardant ADP/PEO Solid Polymer Electrolyte for Dendrite-Free and Long-Life Lithium Battery by Generating Al, P-rich SEI Layer. *Nano Lett* **21**, 4447-4453 (2021).
28. Wang, H. *et al.* Thiol-Branched Solid Polymer Electrolyte Featuring High Strength, Toughness, and Lithium Ionic Conductivity for Lithium-Metal Batteries. *Adv Mater* **32**, e2001259 (2020).
29. Yang, X. *et al.* Determining the limiting factor of the electrochemical stability window for PEO-based solid polymer electrolytes: main chain or terminal -OH group? *Energy & Environmental Science* **13**, 1318-1325 (2020).
30. Cui, Y. *et al.* A Fireproof, Lightweight, Polymer-Polymer Solid-State Electrolyte for Safe Lithium Batteries. *Nano Lett* **20**, 1686-1692 (2020).
31. He, Y. *et al.* Stereolithography Three-Dimensional Printing Solid Polymer Electrolytes for All-Solid-State Lithium Metal Batteries. *Nano Lett* **20**, 7136-7143 (2020).
32. Li, X. *et al.* Designing Comb-Chain Crosslinker-Based Solid Polymer Electrolytes for Additive-Free All-Solid-State Lithium Metal Batteries. *Nano Lett* **20**, 6914-6921 (2020).
33. Pei, F. *et al.* Interfacial self-healing polymer electrolytes for long-cycle solid-state lithium-sulfur batteries. *Nat Commun* **15**, 351 (2024).
34. Wu, L. *et al.* Flame-Retardant Polyurethane-Based Solid-State Polymer Electrolytes Enabled by Covalent Bonding for Lithium Metal Batteries. *Advanced Functional Materials* **34** (2023).
35. Lin, X. *et al.* A self-healing polymerized-ionic-liquid-based polymer electrolyte enables a long lifespan and dendrite-free solid-state Li metal batteries at room temperature. *Mater Horiz* **10**, 859-868 (2023).
36. Daniels, E. L., Runge, J. R., Oshinowo, M., Leese, H. S. & Buchard, A. Cross-Linking of Sugar-Derived Polyethers and Boronic Acids for Renewable, Self-Healing, and Single-Ion Conducting Organogel Polymer Electrolytes. *ACS Appl Energy Mater* **6**, 2924-2935 (2023).
37. Ling, C. *et al.* In-situ polymerization induced phase separation to develop high-performance self-healable polymeric electrolytes for lithium metal battery. *Materials Today Energy* **36** (2023).
38. Zhao, L. *et al.* Dynamic Supramolecular Polymer Electrolyte to Boost Ion Transport Kinetics and Interfacial Stability for Solid-State Batteries. *Advanced Functional Materials* **33** (2023).
39. Huang, Y., Shi, Z., Wang, H., Wang, J. & Xue, Z. Shape-memory and self-healing polyurethane-based solid polymer electrolytes constructed from polycaprolactone segment and disulfide metathesis. *Energy Storage Materials* **51**, 1-10 (2022).
40. Chang, C. *et al.* Self-healing single-ion-conductive artificial polymeric solid electrolyte interphases for stable lithium metal anodes. *Nano Energy* **93** (2022).
41. Jaumaux, P. *et al.* Deep-Eutectic-Solvent-Based Self-Healing Polymer Electrolyte for Safe and Long-Life Lithium-Metal Batteries. *Angew Chem Int Ed Engl* **59**, 9134-9142 (2020).
42. Liu, Q. *et al.* Self-Healing Janus Interfaces for High-Performance LAGP-Based Lithium Metal Batteries. *ACS Energy Letters* **5**, 1456-1464 (2020).
43. Zhou, B. *et al.* Flexible, Self-Healing, and Fire-Resistant Polymer Electrolytes Fabricated via Photopolymerization for All-Solid-State Lithium Metal Batteries. *ACS Macro Lett* **9**, 525-532 (2020).
44. Jing, B. B. & Evans, C. M. Catalyst-Free Dynamic Networks for Recyclable, Self-Healing Solid Polymer Electrolytes. *J Am Chem Soc* **141**, 18932-18937 (2019).
45. Ye, F. *et al.* High-Entropy Polymer Electrolytes Derived from Multivalent Polymeric Ligands for Solid-State Lithium Metal Batteries with Accelerated Li(+) Transport. *Nano Lett* **24**, 6850-6857 (2024).

46. Zhao, Y. *et al.* Opening and Constructing Stable Lithium-ion Channels within Polymer Electrolytes. *Angew Chem Int Ed Engl* **63**, e202404728 (2024).
47. Tang, L. *et al.* Polyfluorinated crosslinker-based solid polymer electrolytes for long-cycling 4.5 V lithium metal batteries. *Nat Commun* **14**, 2301 (2023).
48. Zhou, H. Y. *et al.* Supramolecular Polymer Ion Conductor with Weakened Li Ion Solvation Enables Room Temperature All-Solid-State Lithium Metal Batteries. *Angew Chem Int Ed Engl* **62**, e202306948 (2023).
49. Xie, X. *et al.* Influencing Factors on Li-ion Conductivity and Interfacial Stability of Solid Polymer Electrolytes, Exemplified by Polycarbonates, Polyoxalates and Polymalonates. *Angew Chem Int Ed Engl* **62**, e202218229 (2023).
50. Zhao, Z. *et al.* Regulating Steric Hindrance of Porous Organic Polymers in Composite Solid-State Electrolytes to Induce the Formation of LiF-Rich SEI in Li-Ion Batteries. *Angew Chem Int Ed Engl* **62**, e202308738 (2023).
51. Qi, S. *et al.* Enabling Scalable Polymer Electrolyte with Dual-Reinforced Stable Interface for 4.5 V Lithium-Metal Batteries. *Adv Mater* **35**, e2304951 (2023).
52. Mu, K. *et al.* Hybrid Crosslinked Solid Polymer Electrolyte via In-Situ Solidification Enables High-Performance Solid-State Lithium Metal Batteries. *Adv Mater* **35**, e2304686 (2023).
53. Dong, X., Mayer, A., Liu, X., Passerini, S. & Bresser, D. Single-Ion Conducting Multi-block Copolymer Electrolyte for Lithium-Metal Batteries with High Mass Loading NCM811 Cathodes. *ACS Energy Letters* **8**, 1114-1121 (2023).
54. Lee, M. J. *et al.* Elastomeric electrolytes for high-energy solid-state lithium batteries. *Nature* **601**, 217-222 (2022).
55. Li, W. *et al.* SnF(2) -Catalyzed Formation of Polymerized Dioxolane as Solid Electrolyte and its Thermal Decomposition Behavior. *Angew Chem Int Ed Engl* **61**, e202114805 (2022).
56. Su, Y. *et al.* High-Entropy Microdomain Interlocking Polymer Electrolytes for Advanced All-Solid-State Battery Chemistries. *Adv Mater* **35**, e2209402 (2023).
57. Xiang, J. *et al.* A flame-retardant polymer electrolyte for high performance lithium metal batteries with an expanded operation temperature. *Energy & Environmental Science* **14**, 3510-3521 (2021).
58. Lin, Z. *et al.* Molecular structure adjustment enhanced anti-oxidation ability of polymer electrolyte for solid-state lithium metal battery. *Nano Energy* **98** (2022).
59. Chen, L. *et al.* In situ construction of Li₃N-enriched interface enabling ultra-stable solid-state LiNi_{0.8}Co_{0.1}Mn_{0.1}O₂/lithium metal batteries. *Nano Energy* **100** (2022).
60. Li, J. *et al.* Constructing interfacial gradient layers and enhancing lithium salt dissolution kinetics for high-rate solid-state batteries. *Nano Energy* **102** (2022).
61. Liang, H. P. *et al.* Polysiloxane-Based Single-Ion Conducting Polymer Blend Electrolyte Comprising Small-Molecule Organic Carbonates for High-Energy and High-Power Lithium-Metal Batteries. *Advanced Energy Materials* **12** (2022).
62. Wang, Y. *et al.* Solid-state rigid-rod polymer composite electrolytes with nanocrystalline lithium ion pathways. *Nat Mater* **20**, 1255-1263 (2021).
63. Yu, X. *et al.* Selectively Wetted Rigid-Flexible Coupling Polymer Electrolyte Enabling Superior Stability and Compatibility of High-Voltage Lithium Metal Batteries. *Advanced Energy Materials* **10** (2020).